# The basis of easy controllability in Boolean networks

Enrico Borriello [1✉] & Bryan C. Daniels [1✉]

Effective control of biological systems can often be achieved through the control of a surprisingly small number of distinct variables. We bring clarity to such results using the formalism of Boolean dynamical networks, analyzing the effectiveness of external control in selecting a desired final state when that state is among the original attractors of the dynamics. Analyzing 49 existing biological network models, we find strong numerical evidence that the average number of nodes that must be forced scales logarithmically with the number of original attractors. This suggests that biological networks may be typically easy to control even when the number of interacting components is large. We provide a theoretical explanation of the scaling by separating controlling nodes into three types: those that act as inputs, those that distinguish among attractors, and any remaining nodes. We further identify characteristics of dynamics that can invalidate this scaling, and speculate about how this relates more broadly to non-biological systems.

[1] School of Complex Adaptive Systems, Arizona State University, Tempe, AZ, USA. ✉email: enrico.borriello@asu.edu; bryan.daniels.1@asu.edu

The development of a comprehensive theory for control of complex biological systems is a major goal of systems biology[1,2]. While many results are available in control theory, its common assumptions have needed to be updated for application in biology: biological processes are typically intrinsically nonlinear[3] and we often care only about coarse-grained output states[4] as opposed to the ability to force a system into any possible state[5].

In the context of Boolean models of gene regulatory networks, a particularly useful concept in the static control of regulatory logic is the control kernel (CK). Introduced in[6], the CK is defined as a minimal set of genes such that external control of their expression is sufficient to steer the network dynamics toward a desired steady gene activation pattern (attractor).

Existing literature on Boolean networks contains a number of related definitions of minimal control sets and efficient approaches to finding them. First, stable motif analysis similarly defines control sets by identifying positive circuits in the network that can self-sustain an associated state (namely, trap spaces in the dynamics)[7]. Individual cyclic attractors are not determined by the method, but grouped into "quasi" attractors that share the same stationary parts. Otherwise, sets of control nodes are defined for quasi-attractors in the same way as CKs. Alternatively, a control set can be defined as a single set of nodes that can force the system to any of the original steady states. Feedback vertex sets, namely minimal sets of nodes whose removal deprives the network of all its directed cycles, efficiently produce an upper bound on minimal control set size that does not require knowing the specific dynamics governing each node[5,8,9]. Finally, the method of differentially expressed positive circuits begins with minimal information about the network structure to efficiently find sets of nodes that, when forced, move the system from one of the network's attractors to another of its attractors[10].

Here, we focus on the CK definition of control, such that a CK particular to one of the original steady states produces that state regardless of the initial condition. Using this definition, and focusing on the particular steady state most often reached when starting from random initial conditions, Kim et al. found empirically in eight representative models that the size of the CK is both uncorrelated with the network size and small[6]. This is in agreement with both the aforementioned existing theoretical approaches and an increasing wealth of empirical findings in cellular reprogramming experiments[11–17], where overexpression of fewer than a dozen transcription factors is capable of selecting a desired phenotypic behavior in systems with tens of thousands of genes. If controlling a network is not an increasingly difficult task as the size of the network increases, then the question arises of why some networks are more amenable than others to external control.

Naively, one might guess that the size of CKs would be close to the logarithm of the number of original attractors in a system. If the possible states of individual nodes are roughly equally likely given the states of other nodes, then forcing one node into a particular state will typically cut the number of attainable states in half. Then by controlling $c$ nodes we expect to be able to narrow $2^c$ possible states down to 1 possible state. Yet the problem of finding the minimal set of controlling nodes is nontrivial, proven to be NP-hard by Akutsu et al.[18], so that no algorithm is expected to run faster in the worst case than checking every possible subset of increasing size. It is worth noting a difference between the approach adopted by Akutsu et al.[18] and Kim et al.[6]. In the first study, new control nodes are added to the original internal nodes of the network. Kim et al. instead actively turn a subset of internal nodes into control nodes. This difference does not alter the computational complexity of the problem.

Here, across 49 example biological networks, we compute CKs for all attractors. We first corroborate earlier results that CKs remain relatively small, finding typical CK sizes smaller than about a dozen nodes even in large networks with up to 72 interacting components.

More importantly, we illuminate the origins of this easy controllability by showing that the average CK sizes do in fact scale logarithmically with the number of attractors. For a number of reasons, this close correlation with the number of attractors has not been highlighted in prior studies. In particular, the scaling is not as clearly evident when looking only at the attractor most often reached from random starting points[6], nor when bounding the control set using the feedback vertex set, which does not require solving for individual attractors[5].

The theoretical justification for the observed scaling is subtle, and we progress here by breaking down the CK problem into three subproblems, one being the witness set problem from discrete mathematics. The lower bound realized by the witness set typically provides a good first approximation to the full size of the CK, which connects the observed logarithmic scaling to known results in computational learning theory.

## Results

**Boolean networks and attractor dynamics**. This section defines common terms and concepts from the study of Boolean dynamical systems. Readers familiar with Boolean network dynamics can proceed to the next section.

Given their abstract nature, the applicability of Boolean networks extends far beyond the mathematical description of biological regulatory networks (cfr.[19–22], to cite just a few examples of a large literature). In the context of theoretical biology, their relevance was first highlighted by Kauffman[23], who identified cell types with the attractors of network dynamics, allowing for their mathematical study. The success of Boolean networks in theoretical biology can be attributed to their simplicity, forming the most parsimonious mathematical representation displaying systemic properties of real biological networks[24,25].

The state of a Boolean network at time $t$ is defined by the state of its $n$ nodes, $x_i(t)$, with $i = 1, …, n$. The Boolean nature of the model means that there are only two possible states for node $i$: ON, which corresponds to $x_i(t) = 1$, or OFF, with $x_i(t) = 0$. When the network describes cell regulation, its nodes are often representative of interacting genes, and $x_i(t)$ is the Boolean approximation of the expression level of gene $i$, with the gene being expressed when $x_i(t) = 1$, and inactive otherwise.

The dynamics of a deterministic Boolean network is defined by $n$ Boolean functions $f_1, …, f_n$ whose input is the $x_1(t), …, x_n(t)$ array, and whose output is the configuration of the network at time $t + 1$:

$$
\begin{aligned}
x_1(t+1) &= f_1(x_1(t), … , x_n(t)) \\
&\cdots \\
x_n(t+1) &= f_n(x_1(t), … , x_n(t)).
\end{aligned}
\tag{1}
$$

The form of these Boolean functions can be determined by experiments[26,27]. This parameterization corresponds to a synchronous update of every node in the network. Asynchronous dynamics are more realistic but less mathematically tractable[28] (see Supplementary Material for a discussion of our approach to asynchronous updating). With $n$ nodes, there are $2^n$ possible network configurations (i.e., gene expression patterns in the case of a genetic network). We will refer to these configurations as states of the network, and to the space of all possible network states as the configuration space. The dynamics of the network is

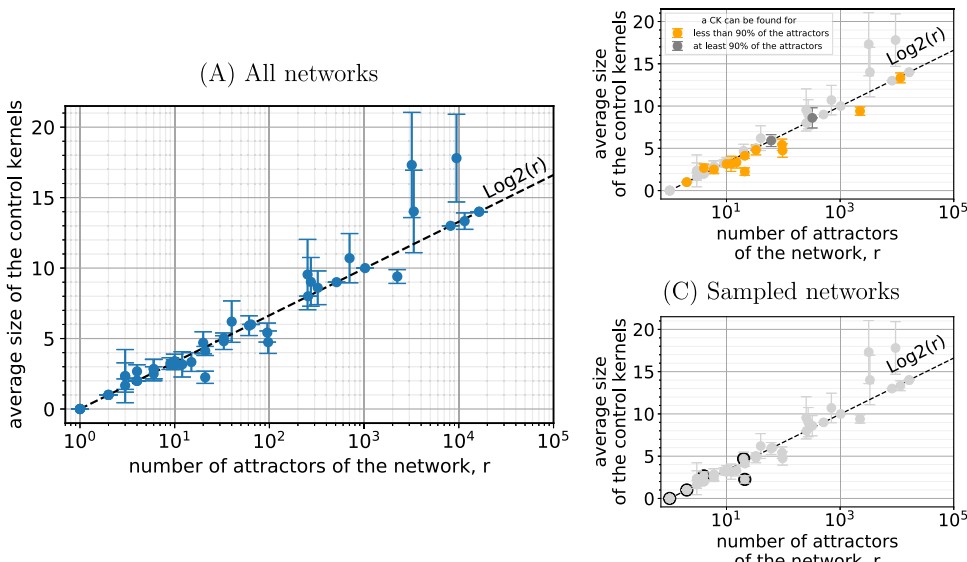

**Fig. 1 Average control kernel size versus the number of attractors in a network. A** The average control kernel size is close to the base-2 logarithm of the total number of attractors. The average is taken over attractors for which a static control kernel exists. Data are presented as mean values ± one standard deviation over attractors (where the number of control kernels contributing is equal to the total number of statically controllable attractors in each case). **B** Some cycles are not statically controllable. **C** Networks analyzed using the sampling method.

represented by a time series of states. Though large, the size of the configuration space is finite. Therefore, whenever the functions $f_i$ are deterministic, these dynamical paths will eventually converge to either a fixed state or a cycle of states. These special sets of states are the attractors of the Boolean network. The set of initial states that converge to a given attractor is known as its basin of attraction. In the case of gene regulatory networks, the core assumption of Kauffman's proposal is the identification between attractors and cell types. Boolean networks can have many distinct attractors, making them ideal for describing how the gene interactions encoded within a single fertilized egg can give rise to a large number of stable cell types[29]. Regardless of their specific interpretation, the attractors of a Boolean network are typically among its most important features, as they correspond to the only possible steady-state dynamics once any transient behavior has ended.

The remainder of this manuscript is devoted to the study of how steady-state dynamics are altered when external control is exerted over the network.

**Logarithmic scaling of control kernel size**. We will assume here that external control is exerted over the network by setting the state of one or more of its nodes to constant values (also referred to as "node state override"[5]). This is done regardless of the updating rules associated to those nodes and the inputs provided by the remaining nodes. This pinning of $p$ nodes is formally a map between dynamical systems, turning a network with $n$ nodes into a "controlled" network with $n-p$ nodes. We define a CK as the minimal amount of pinning necessary to force the system into a desired attractor:

**Definition 1** (CK): For a given attractor **A**, a CK is defined as a set of nodes of minimal order whose pinning reshapes the dynamics such that the basin of attraction of **A** becomes the entire configuration space.

By this definition, for a given **A** multiple CKs of the same size may exist, or there may be no existing CK. The most naive approach to finding a CK for a desired attractor **A** is simple but

time-consuming: pin every possible subset of nodes of increasing size to their values in **A** and test the network's dynamics to determine whether all initial states of the network converge to **A**. In the case of networks with highly modular organization, it is possible to be much more efficient by computing CKs for each module separately (see "Methods").

For every fixed-point attractor, at least one CK is guaranteed to exist: at worst, we can succeed in control by pinning all nodes to their values in the desired attractor. Cyclic attractors do not necessarily have a CK. We know we cannot pin any nodes that do not have a constant value in the attractor, as this will not be consistent with the original attractor. This limitation comes from our assumption of static pinning in Definition 1. Generalizations that use dynamic pinning are also possible, which we do not explore in any detail here. Such forms of dynamic control seem less relevant from a biological perspective and are more difficult to enforce experimentally.

To determine the typical sizes of CKs in biological examples, we analyzed models in the Cell Collective database[30]. They span, but they are not limited to, biological cellular processes including cycles, differentiation, plasticity, migration, and apoptosis. They exemplify biological processes in humans and other animals, as well as plants, bacteria, and viruses. For 44 networks, using a reasonable amount of computing time (roughly 1 processor running for 1 day for each network), our algorithm was able to find CKs for all attractors for which a CK exists. For an additional five networks for which we were unable to exhaustively find all attractors, we computed CKs for a set of attractors reached from $10^4$ to $10^6$ random initial conditions (see "Methods"). The networks we analyzed range in size from 5 to 73 nodes (see "Methods"). In computing CKs for all known attractors of each network, we depart from the approach adopted in[6], which studies only the CK for the attractor with the largest basin.

We find empirically that the average CK size $\langle|CK|\rangle$ in each network is close to the base-2 logarithm of the total number of attractors $r$ (Fig. 1). The reasons for this scaling are subtle, and in the next section we propose a theoretical interpretation and speculate about whether this scaling persists at larger $n$.

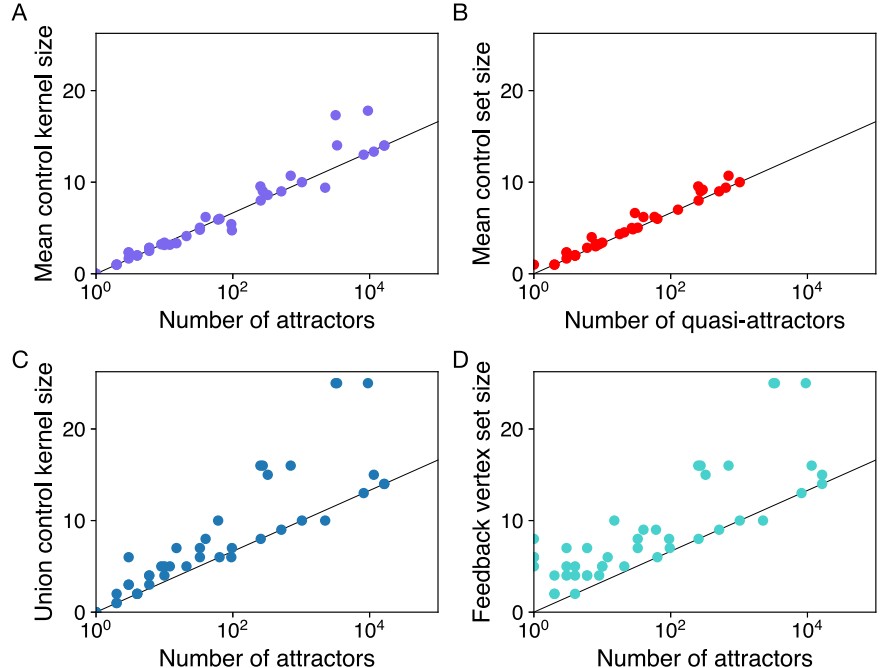

**Fig. 2 Comparing our control kernel results with those obtained using two alternative methods for computing controlling nodes. A** For comparison, we replot mean control kernel size versus number of attractors, as in Fig. 1A. **B** Mean control set sizes computed for 36 networks using the stable motif algorithm, which groups some cyclic attractors into quasi-attractors. Logarithmic scaling of the control set sizes with respect to the number of quasi-attractors is clearly visible. **C** For comparison to the feedback vertex set, which places an upper bound on the minimal control set required to reach all attractors, we compute the size of the union of control kernel nodes of each network across all its attractors. **D** The size of feedback vertex sets is comparable to union control kernel sizes in **C**.

The observed trend is a potential explanation for why models of biological networks have small CKs. Biological CKs may remain small regardless of the size of the network because the necessary control depends mainly on the logarithm of the number of possible outcomes, which may be fundamentally limited in functional networks.

**Comparing our results with those obtained using alternative methods**. In Fig. 2, we compare our CK size results to two other methods: stable motifs[7] and feedback vertex sets[5,8,9].

First, the method of stable motifs groups cyclic attractors with identical stationary parts into quasi-attractors (see "Introduction" and[7]), meaning that controlling for individual cycles is not necessarily possible in all cases. Still, we see in Fig. 2B very similar scaling results for stable motif control set sizes as a function of the number of quasi-attractors. Note that, using a similar amount of computing time as for our CKs and without further optimization, we were able to compute stable motif control sets for only 36 biological networks.

Second, we expect feedback vertex sets to provide an upper bound on the size of the single set of control nodes needed to reach all attractors[5]. A set achieving an analogous goal, but still defined in terms of attractor-specific CKs, is a "union" CK, namely a set of nodes that includes a CK for each attractor. Note that we do not attempt to find the union CK of minimal size, which could be smaller in cases in which multiple possible CKs exist for individual attractors. Plotting the sizes of the union of CKs across attractors for each network, we see similar results to the sizes of feedback vertex sets (Fig. 2C, D). Note, too, that logarithmic scaling of the feedback vertex set size is much less pronounced than in the case of attractor-specific CKs in Fig. 2A.

Additionally, we analyze the overlap between the specific nodes in our CKs and the controlling nodes identified by the other two methods. Note that the existence of multiple possible minimal control sets and the NP-complete nature of the problem limits us to compare representative sets of minimal size. Comparing CKs to stable motif control sets (for fixed points only), the average overlap across all 4262 attractors is 94%. When comparing union CKs to feedback vertex sets, the average proportion of CK nodes that are part of one particular feedback vertex set is 88%. (For more details, see Fig. S5 in the Supplementary material.)

**Road map of the theoretical interpretation**. In the remainder of the Results, we investigate the conceptual reasons for the scaling behavior in Fig. 1, connecting it to an unsolved problem in computational learning theory. What follows is a schematic guideline of the steps that lead us to the result:

- We start by identifying three contributions to the CK, with different properties: input nodes, distinguishing nodes, and additional nodes. We then propose a method for bounding the CK size that naturally provides the controlling nodes in this order.
- To find CKs for a network, the global dynamics must generally be re-solved for multiple combinations of pinned nodes of increasing size. We show that the additional nodes are the only nodes whose identification requires this repeated solving. Identifying input nodes is trivial, and distinguishing nodes require knowing only the network's original attractors and solving a problem equivalent to finding minimal order witness sets.
- We then show that the additional nodes provide a subdominant contribution to the size of the CK for the networks we study. By neglecting the additional nodes, we exhibit a "first-order approximation" of the CK that we compare to the full calculation in Fig. 1. We show that the approximation scales logarithmically with the number of attractors $r$, thus explaining our main result.

- We also discuss cases that may significantly deviate from this scaling behavior. We identify two possible scenarios: (1) Cases where the contribution of the additional nodes is dominant; (2) cases where the average size of the minimal witness sets exceeds $\log_2 r$.

**Input nodes**. Let us start by considering a special class of nodes that are easy to analyze separately: those that have a constant value in time. In biological models, there are often examples of nodes whose dynamics is described by the identity function $x_p(t + 1) = x_p(t)$, as it offers a simple implementation of external inputs acting on the network. Input nodes appear often in biological networks as either signals from other subsystems or as external variables controlled in experiments (e.g.[31]). With this in mind, we will assume the following definition:

**Definition 2** (Input node): *A node is an input node if its updating rule is the identity function.*

Having input nodes increases the number of attractors in a network, as each setting of the inputs corresponds to at least one distinct attractor.

In steering the dynamics toward a given attractor, we are forced to pin the input nodes to the values they take in the attractor. Otherwise, both values for each input would remain viable, and we cannot select between them by controlling the other nodes. As this step is unavoidable we can easily conclude the following:

**Proposition 1**: *A CK must include all the input nodes of the network.*

Therefore, we pin all input nodes as the initial step of our analysis. When present, input nodes act as control nodes as defined in[18], and in agreement with the findings of that study. Note that it is possible for nodes that were not input nodes in the initial network to become effective input nodes after this pinning procedure. A simple example is a node whose state depends only on input nodes. We do not distinguish this situation in our study. Therefore, these nodes are counted among the distinguishing nodes as defined in the next subsection.

**Distinguishing nodes**. Let us now examine the effect of the pinning procedure on the state transitions. It is easier to analyze the changes in terms of the transition matrix, the table of correspondences between the state at time $t$, $x = (x_1, ..., x_n)$ (the left side of the table) and the subsequent state at time $t + 1$, $x' = f(x) = (x'_1, ..., x'_n)$ (the right side):

| state at time $t$ | | | | state at time at time $t + 1$ | | | |
|---|---|---|---|---|---|---|---|
| $x_1$ | ... | $x_{n-1}$ | $x_n$ | $x'_1$ | ... | $x'_{n-1}$ | $x'_n$ |
| 0 | ... | 0 | 1 | ... | ... | ... | ... |
| 0 | ... | 1 | 0 | ... | ... | ... | ... |
| ... | ... | ... | ... | ... | ... | ... | ... |
| 1 | ... | 1 | 1 | ... | ... | ... | ... |

Pinning node $j$ to 0 (or 1) corresponds to deleting column $x_j$ and $x'_j$ and deleting the $2^{n-1}$ rows with the opposite value $x_j = 1$ (or 0). Note that a fixed-point attractor corresponds to a row in the transition matrix with the left side equal to the right side (i.e., $x_k = x'_k$ for all $k$ in that row). Pinning a node will remove a preexisting attractor if, and only if, it appears on the left side of a row we are eliminating. When pinning to control the network into a desired attractor, we must at least eliminate all other initial attractors:

**Definition 3** (Distinguishing nodes): *A subset of nodes for which a pinning exists that is both compatible with attractor A*

and incompatible with the other initial attractors of the network is a set of distinguishing nodes of **A**.

In order to remove all attractors other than attractor **A**, we need to pin a set of its distinguishing nodes, which leads us to the following:

**Proposition 2**: *A CK of attractor A must include a set of distinguishing nodes for attractor A with respect to the network's original attractors.*

Therefore, a more efficient and still exact method for finding CKs is to check not every possible subset of nodes, but to check only distinguishing node sets (since we know that the CK must include at least one of them). This is the method we use to obtain our results in Fig. 1.

We now have a lower bound on the size of the CK. To state this precisely, let us consider a network with $n$ nodes, $r$ attractors, and $m$ input nodes. Let us then select an attractor **A** and pin all input nodes to values compatible with **A**. The new network (with initial input nodes pinned) will possess a certain number $r'$ of attractors: **A** plus other attractors $\mathbf{B}_1, ..., \mathbf{B}_{r'-1}$. Let us now call $w^{(1)}$ any minimal order set of nodes distinguishing **A** from $\mathbf{B}_1, ..., \mathbf{B}_{r'-1}$. Then the size of the CK is bounded below by $m + |w^{(1)}|$ (Propositions 1 and 2).

**Additional control nodes**. The reason that input and distinguishing nodes provide only a lower bound to CK size is that the pinning intervention can also create new attractors. For instance, we get a new fixed-point attractor whenever the left and right sides of a row in the transition matrix are equal over the $n - 1$ remaining columns, with differing values only in the column that we are eliminating. Note that pinning an input node cannot create new attractors. (In the row we are eliminating, input nodes by definition have equal values on the left and right sides.)

Pinning the nodes in $w^{(1)}$ will remove $\mathbf{B}_1, ..., \mathbf{B}_{r'-1}$, but this may now create new attractors. We can also get an upper bound by iteratively pinning subsequent minimal distinguishing node sets that are needed to remove these additional attractors:

$$m + |w^{(1)}| \le |\text{CK}| \le m + |w^{(1)}| + \sum_{i \ge 2} |w^{(i)}|, \qquad (2)$$

where $w^{(i)}$ with $i \ge 2$ are the sets of additional nodes we need to pin after we have already removed $\mathbf{B}_1, ..., \mathbf{B}_{r'-1}$. This iterative procedure produces only a bound instead of the exact CK size because iterative pinning can in some cases mislead us toward a pinning that works but is not minimal. To mitigate this aspect, in performing the iterative procedure, we choose among different minimal sets of distinguishing nodes by picking the one that creates the fewest new attractors. We find numerically that the bound computed in this way is often tight (see Fig. S7 in the Supplementary material).

Also notice that finding the distinguishing nodes $w^{(1)}$ only requires knowledge of the initial attractors. On the contrary, to find the additional control nodes we need to distinguish the selected attractors from new attractors created during each round of pinning. This requires solving the global dynamics of the network for each combination of pinning nodes.

**Relative contributions of the successive rounds of pinning**. In the following, we will refer to the pinning of the input nodes as round zero in this iterative procedure. Pinning $w^{(1)}$ is what we will call the first round of pinning, while any additional control nodes will be pinned during additional rounds of pinning. We summarize this procedure in Fig. 3, where we show the process determining the CK of the attractor with smallest basin size in a gene regulatory network underlying early cardiac development[32].

Despite the iterative nature of our pinning procedure, these three steps possess very different characteristics. Round zero is

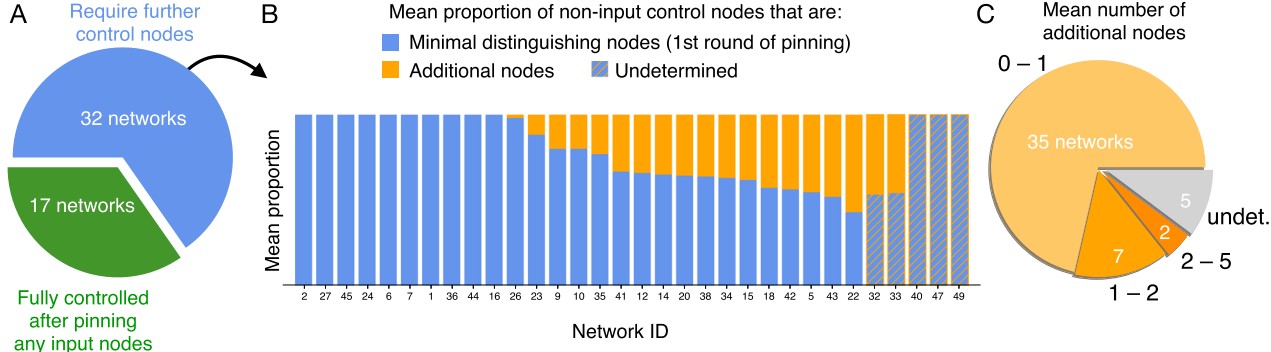

**Fig. 3 Schematic of our iterative pinning procedure.** This example demonstrates selecting a particular attractor in the network describing the core gene regulatory network for early cardiac development[32]. (Note that we use the iterative procedure only to put bounds on the control kernel size as in Eq. (2), while the true minimal control kernel can be computed by an exhaustive procedure as explained in the Methods. In this particular case, the two methods produce identical results.).

**Fig. 4 Relative contributions of the successive rounds of pinning.** After accounting for input nodes, minimal distinguishing node sets $w^{(1)}$ account for much of the size of control kernels in the 49 biological networks. **A** Some networks are fully controlled after pinning only input nodes, but most require further pinning. **B** Vertical bars indicate the mean proportion of non-input control nodes that correspond to minimal distinguishing nodes (blue) and those that must be set by additional pinning (orange). Network IDs correspond to those listed in Table 1. Some cases remain "undetermined" as identifying the minimal size of distinguishing node sets can be computationally infeasible (see "Methods"). **C** In most networks, the mean number of additional nodes needed beyond the minimal distinguishing nodes remains small.

trivial, and it does not introduce new attractors. The first and the subsequent rounds of pinning represent NP-hard problems, and they can introduce new attractors.

Remarkably, in our data set the first round typically identifies most of the necessary control nodes (Fig. 4). For this reason, we will call the input and distinguishing nodes a first-order approximation of the CK, neglecting any additional control nodes. To understand why we expect this approximation to be reasonable, let us examine the likelihood of introducing new attractors through pinning.

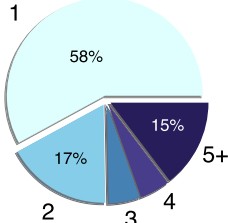

Number of attractors
remaining after pinning

**Fig. 5 The number of attractors remaining after each iterative pinning is typically small.** Here we use the iterative pinning procedure to find upper bounds on control kernel sizes for 40 networks that we can analyze using the sampling method. This results in a total of 4484 iterative rounds. Having 1 attractor remaining after pinning indicates success in finding a controlling set.

**Fixed-point attractors**. With certain assumptions about network properties, it can be possible to estimate the expected number of fixed-point attractors. This is the case, for example, in a well-studied class of dynamical systems known as random Boolean networks (RBNs)[23]. RBNs are defined in terms of a generative mechanism for both their topology and dynamical rules, and this can be used to prove that, on average, a given RBN possesses just one fixed-point attractor[33].

The networks in our database are not random networks. They were instead inferred from data by a number of research groups. Interestingly, after the input and distinguishing nodes have been pinned the number of additional attractors is also typically small (Fig. 5). A heuristic argument that we detail below reveals general conditions under which we expect this to be true.

The argument is very simple: Whatever states appear on the right side of the transition matrix, we have an even distribution of 0s and 1s on the left side. Without knowledge of the specific mapping of initial states to next states, our best assumption is that they appear in random order on the right side, leaving unspecified which initial state corresponds to each next state. For each row, the probability of having the right side matching the left side entry by entry is $1/2^{n'}$, where $n'$ is the current number of nodes we have not pinned. We have $2^{n'}$ rows. Therefore, the expected number of fixed-point attractors would be 1. Given the heuristic nature of this argument, we cannot assume this to be more accurate than just an order of magnitude estimate. One reason why this number could be significantly greater than 1 would be the presence of many input nodes (which would guarantee the existence of at least $2^m$ attractors), but we have already removed all the initial input nodes in round zero. In this way, we have reduced the study of the controllability of each of our networks to the study of a set of networks without input nodes. Notice that RBNs do not have input nodes, unless that happens by accident. Pinning the input nodes makes our networks acquire one of the main topological features of RBNs.

**Cyclic attractors**. We have neglected cycles in our previous argument. A cycle over states $i$ and $j$ would require the simultaneous conditions (1) left side of row $i$ = right side of row $j$ and (2) left side of row $j$ = right side of row $i$. Proportionally longer chains of conditions would be required for longer cycles. In the case of RBNs, the intuitive idea that longer cycles become less likely due to the multiple conditions they require holds only for relatively small networks, e.g., networks with fewer than about 20 nodes (cfr. Fig. 1 of[33]). This intuition inevitably breaks down as the network size increases, due to the combinatorial growth in the

number of ways a cycle of states can be closed. Eventually, cycles become dominant for RBNs with more than about 300 nodes.

A proliferation of cycles created during the intermediate rounds of pinning could potentially prevent our iterative procedure from converging quickly. Nonetheless, this is not something we observe in the biological networks we analyzed. Figure 5 shows that the number of attractors remaining after each iterative pinning, including cycles, is typically small. Interestingly, our networks fall within the size range for which RBNs would present a number of cycles comparable to the number of fixed points. Whether biological networks much larger than the sizes we analyzed display a proliferation of cycles similar to RBNs is something we cannot currently test. Indeed, it is an open question whether networks displaying large numbers of cycles are relevant to biology[34] and complexity science more generally. It is also worth noting that many cycles in large networks become unstable when the network state is updated in a nondeterministic asynchronous way[35] (see Supplementary material).

Our brief discussions about fixed-point and cyclic attractors reveal that—once the input nodes are pinned, and in the absence of additional structure in the dynamics that significantly undermine the approximate randomness of the transition matrix (for an example of this see the section on Random Networks)—we can expect our iterative pinning procedure to converge quickly. With a small number of attractors remaining after each round of pinning, there is a fairly high probability of having, at some step, only one attractor, at which point a CK has been found and iteration stops. This probability, which we empirically observe to be 58% across the networks we study (Fig. 5), confirms our simple interpretation.

We have anticipated that our first-order approximation of the $|CK|$ consists of neglecting the additional nodes (the lower bound in Eq. (2)). A fast convergence of the pinning procedure guarantees a small number of rounds of pinning (the number of nonzero terms $|w^{(i)}|$ with $i \geq 2$ in Eq. (2)), but it says nothing about the size of those terms, and therefore does not guarantee that additional nodes cannot dominate $|CK|$. Therefore, to rule out this possibility, we need to better understand the expected size of minimal distinguishing node sets.

**Witness sets**. What we call a set of distinguishing nodes of an attractor is closely related to a known concept in discrete mathematics, the witness set[36]. Witness sets are a recurring topic in computational learning theory[37] and are referred to by multiple names, including discriminants[38] and specifying sets[39]. Another related quantity is the teaching dimension of set $\mathcal{A}$[40]: While we are interested in the average size of the smallest witness sets $\bar{w}$, the teaching dimension of $\mathcal{A}$ is the size of its largest minimal $w_i$.

Given a list of binary vectors, a witness set for one of the vectors consists of bits that, when revealed, distinguish it from all other vectors in the list. Formally:

**Definition 4** (Witness set): Let $\mathcal{A} \subseteq \{0,1\}^n$ be a family of $r$ distinct binary $n$-tuples of length $n$. A set $W_i \subseteq \{1, 2, \ldots, n\}$ of coordinates is a witness set for the $n$-tuple $\mathbf{A}_i \in \mathcal{A}$ if for every other $\mathbf{A}_j \in \mathcal{A}$ there exists a coordinate $x^{[j]}$ in $W_i$ such that $x^{[j]}$ in $A_i$ differs from $x^{[j]}$ in $A_j$.

When all attractors are fixed points, we define $\mathcal{A}$ as the set of the $r$ attractors $\mathbf{A}_i$. The witness set of minimal size, $w_i$, for a given attractor is then equivalent to the minimal set of distinguishing nodes $w^{(1)}$ defined above.

Cyclic attractors add two complications to this correspondence between witness sets and distinguishing nodes, but the basic idea remains the same. First, when distinguishing an attractor $\mathbf{A}_i$ from a cycle, we treat nodes whose values change within the cycle as

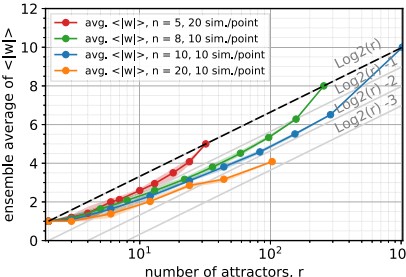

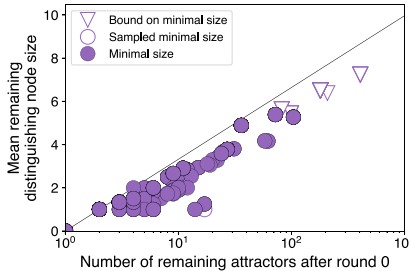

**Fig. 6 Average ⟨|w|⟩ values for increasing length n of random n-tuples.** Each point is averaged over random realizations of $\mathcal{A}$. The shaded region around each curve marks the minimum and the maximum averages we find. $\log_2 r$ looks like an effective upper bound for not just the average ⟨|w|⟩, but also the individual values of ⟨|w|⟩ for all choices of n, r, and the number of simulations we consider. Remarkably, ⟨|w|⟩ never departs from $\log_2 r$ by more than a few units.

**Fig. 7 Distinguishing node set sizes versus number of remaining attractors.** Mean minimal distinguishing node set sizes $\langle |w_j^{(1)}| \rangle$ are empirically less than or equal to $\log_2 r_j$ in all biological cases we analyze. This consists of 43,289 cases (with 12 sampled cases).

always being distinguished from a fixed value in $\mathbf{A}_i$. This can be accomplished, for instance, by treating each non-constant node in the cycle as if it had the opposite value of the one in $\mathbf{A}_i$. Second, when distinguishing a cycle from other attractors, we first restrict the "columns" of $\mathcal{A}$ to include only those nodes that have constant values in the cycle, as we assume that distinguishing nodes must be pinned to static values. We have already seen (cfr. the comments following our Definition 1) that a witness set might not exist in this case.

The key question then becomes: What sets the typical size of minimal witness sets $w_i$? Simple examples show that $|w_i|$ can easily be as large as n for an individual i—for instance, if n other $\mathbf{A}_j$ are chosen that differ from $\mathbf{A}_i$ by one bit flip for each of the n nodes. However, creating an example with large $|w_i|$ for all i requires larger r. The more relevant question then becomes: what sets the average witness set size over attractors ⟨|w|⟩ as a function of the number of attractors r?

Naively, one might expect that ⟨|w|⟩ scales as $\log_2 r$: On average, revealing a given bit will distinguish the desired attractor from about half of the remaining attractors, so one would expect to be able to distinguish any given attractor using about $\log_2 r$ bits.

Known topological features exist that naturally produce a logarithmic scaling of the number of distinguishing nodes. For example, we have already seen that a large number of input nodes in the network would induce logarithmic scaling. This also holds when the network possesses disconnected components, though this is not the most relevant case in biology. (Non-interacting modules have a multiplicative number of attractors and an additive number of minimal distinguishing nodes, which leads to logarithmic scaling.) To a lesser extent a hierarchical structure of interacting modules would favor the same scaling. Input nodes and modularity are important features of our database, but further analysis indicates they are not solely responsible for the $\log_2 r$ scaling (see "Methods").

Even in the absence of such topological features, numerical experiments show that $\log_2 r$ is often an apparent upper bound on ⟨|w|⟩, and this was long conjectured to be an exact result[36]. Yet definite exceptions to this apparent bound are now known to exist, and one such exception will be detailed in one of the next sections (see "Finite projective plane"). We first discuss a numeric experiment demonstrating that logarithmic scaling is indeed a typical scenario.

We start by randomly selecting r Boolean vectors and assuming that they constitute the set of attractors $\mathcal{A}$. This way we leave the topology of the hypothetical network generating them unspecified. The result is shown in Fig. 6, where average ⟨|w|⟩ values are

plotted for increasing values of n, each point averaged over realizations of $\mathcal{A}$. The maximum values found for ⟨|w|⟩ always remain below the $\log_2 r$ reference line. The average ⟨|w|⟩ points fall along seemingly convex curves—anchored to the $\log_2 r$ line at $r = 2$ and $r = 2^n$, as expected—and depart from the $\log_2 r$ line by just a few units. The NP-hard nature of finding w limits our ability to simultaneously increase n and r in this analysis. (We can analyze larger networks exactly in Fig. 1 because of their modular nature. We are now intentionally forcing ourselves to deal with the opposite scenario.)

Given this numerical evidence, we have reason to expect that witness set sizes, and therefore distinguishing node sets, are typically no larger than the logarithm of the number of attractors they distinguish. In each round of pinning after the first, the expectation that we have only a few remaining attractors then translates into small numbers of additional nodes added in each round (small terms $|w^{(i)}|$ in Eq. (2) when $i \geq 2$). This, combined with the expectation that only a few rounds contribute, leads us to consider an approximation in which we neglect the additional nodes.

**First-order approximation**. Neglecting the additional nodes provides a "first-order" approximation of the CK:

$$|\text{CK}| \approx |\text{CK}^{(1)}| \equiv m + \langle |w^{(1)}| \rangle. \quad (3)$$

We find empirically that the minimal number of distinguishing nodes is always less than the logarithm of the number of attractors being distinguished (Fig. 7). That is, for the cases we test:

$$\langle |w_j^{(1)}| \rangle \leq \log_2 r_j, \quad (4)$$

where $r_j$ is the number of attractors sharing the jth input configuration ($j = 1, ..., 2^m$). When (4) holds, we can additionally bound the average minimal distinguishing node size $\langle |w^{(1)}| \rangle$ over all $r = \sum r_j$ attractors of a given network (see "Methods"): $\langle |w^{(1)}| \rangle \leq \log_2 r$. Combined with the fact that $m \leq \log_2 r$, we obtain the following bound on the average first-order approximation:

$$\langle |\text{CK}^{(1)}| \rangle \leq 2\log_2 r. \quad (5)$$

We validate this for our cases in Fig. 8, finding in fact that $\langle |\text{CK}^{(1)}| \rangle$ is bounded more strongly by $\log_2 r$ (see "Methods" for further discussion). The bound (5) relies on the empirically validated inequality (4), the more general validity of which we explore in the next section.

We can now summarize to draw our main conclusion. Our iterative pinning procedure adds only a small number of additional nodes after the first round, with the combination of input nodes and minimal distinguishing nodes $w^{(1)}$ representing the most sizable contribution to the CK (Fig. 4). Within the set of biological networks we analyze, this first-order approximation

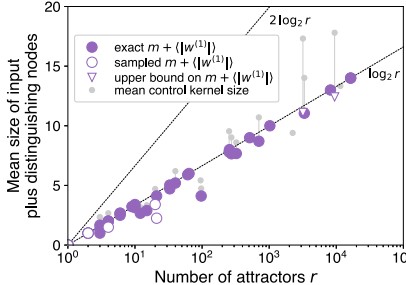

**Fig. 8 First-order approximation of control kernel sizes.** The first-order approximation of control kernel size given by Eq. (3) is empirically less than or equal to $\log_2 r$ in all biological cases we analyze.

(Eq. (3), displayed in Fig. 8) accounts for the logarithmic scaling that we observe in Fig. 1. Furthermore, an apparent bound (5) on this approximation suggests that the logarithmic scaling of control may occur more generally.

**Exceptions**. The logic leading to the logarithmic scaling of $\langle|\mathrm{CK}|\rangle$ could be invalid in two general cases:

- Minimal distinguishing node sets may not be bound by the logarithm of the number of attractors they distinguish (inequality (4) may not hold).
- Additional nodes may make a non-negligible contribution to the CK (approximate equality (3) may not hold).

Though we do not find either of these exceptions among the biological cases we analyzed, we explore a larger set of cases in these final sections to better understand the degree to which exceptions might be expected to arise in other real-world situations.

First, we describe a known construction that produces an exception to inequality (4). We then run our CK analysis on three ensembles of random networks to search for exceptions to logarithmic scaling.

**Finite projective plane**. If universal, the trend shown in Fig. 6 would explain the logarithmic scaling of $\langle|\mathrm{CK}^{(1)}|\rangle$. But at least one exception exists. We owe its ingenious proof to Kushilevitz et al.[36], and it involves the geometry of a finite projective plane.

A finite projective plane[37] of order $p$ ($p$ being a prime number) is a set of $q = p^2 + p + 1$ points and $q$ lines with the following properties:

(1) Any two points determine a line.
(2) Any two lines determine a point.
(3) Every point has $p + 1$ lines on it.
(4) Every line contains $p + 1$ points.

Kushilevitz et al.[36] used this construction to exhibit a set of vectors with $\langle|w|\rangle$ exceeding $\log_2 r$ for large enough $r$. These vectors are chosen to be the rows of the incidence matrix of such a plane (line vectors, or just lines in what follows) plus the rows of the identity matrix of order $q$ (point vectors, or just points). Therefore, $r = 2q$ in this example.

The proof proceeds as follows: to distinguish a point from the other points we just need to pin its only nonzero coordinate to 1. To distinguish it from the lines we first notice that this point is on $p + 1$ lines. Each line contains $p$ more points. Therefore, for each line, we choose a point in it (different from the one we want to select), and pin its coordinate to 0. This way we pin $p + 1$ more coordinates. Therefore, to distinguish a point from the remaining vectors, we need to pin $p + 2$ coordinates.

Distinguishing a line is easier. We just pin the coordinates of 2 of its points to 1. In fact, no point has 2 nonzero coordinates, and no other line contains both those points.

We can now easily calculate $\langle|w|\rangle$ for this set of vectors:

$$\langle|w|\rangle = \frac{2q + q(p+2)}{2q} = \frac{p+4}{2}, \qquad (6)$$

where $p = (\sqrt{2r-3} - 1)/2$. As $p \sim \sqrt{r}$, this construction gives rise to a polynomial scaling of witness set sizes with respect to $r$, and consequently to values of $\langle|w|\rangle$ far exceeding $\log_2 r$ when $r$ is sufficiently large. The first prime number for which this happens is $p = 17$, corresponding to $r = 614$ attractors and a network with $n = 307$ nodes. (More precisely, $\langle|w|\rangle$ can be greater than $\log_2 r$ for $r < 6$ and $r \geq 369$. But the smallest order, $p = 2$, already corresponds to $r = 14$. Therefore, we are only interested in the large $r$ limit.)

As relevant as the existence of this example is the likelihood of analogous cases. One simple test to perform is to check whether flipping a single entry in one of the vectors is enough to bring $\langle|w|\rangle$ below $\log_2 r$. We have explicitly checked this, but in the worst case a bit flip perturbation instead adds to $\langle|w|\rangle$ (see Supplementary material for a more thorough analysis).

It is important to stress that—when interpreted as the attractors of a network—these projective plane exceptions correspond to extremely biased scenarios, where each variable has a total ratio of 1s to 0s over the total number of attractor states equal to $(p+2)/2q \sim 1/\sqrt{r}$, i.e., already as low as 3% when $r = 614$. They constitute an infinitesimal part of the configuration space of all possible sets of $r$ vectors. The likelihood of sets with $\langle|w|\rangle > \log_2 r$ within this space (see[36] for some discussion of this problem), as well as the relevance in natural sciences of networks generating them as steady-states, remain open questions.

**Random networks**. To briefly explore to extent to which we expect exceptions to scaling in more general contexts, we apply our methods to three ensembles of randomly generated networks. First, we use the well-studied ensemble of p–K random networks[41]. In this ensemble, each node receives input from exactly $K = 2$ nodes, and Boolean logic functions are chosen such that the probability of the ON or 1 state appearing on the right-hand side of a given row of the truth table is $p$, a parameter that we vary. Second, we use another ensemble common in the literature that assigns each node a threshold to activation and a set of input nodes whose states are summed and compared to the threshold, with some input nodes acting to excite and some to inhibit[6,42]. Following ref. [6], we set thresholds to zero and vary both the density of interactions as well as the relative proportion of inhibitory edges. In cases with little inhibition, networks are biased toward activation. To explore the effects of this bias, we also include an ensemble of "balanced" threshold networks, in which the thresholds of individual nodes are set not to zero but to the average of incoming edges, such that the distribution of summed inputs is always centered at the threshold.

The results of the CK analysis for networks sampled from these random ensembles are shown in Figs. 9 and 10. Similar logarithmic scaling as in the biological networks is demonstrated by most of the random networks, with a few exceptions of large CK in the case of zero threshold networks (azure triangles in Fig. 9A). These exceptions can be understood by considering the effect of a strong bias toward activation on the set of fixed-point attractors: those fixed points with few or no activated nodes are expected to have small basins because most perturbations will lead toward states with more excited nodes. In particular, the state with all inactivated nodes, which is a fixed point for all zero threshold networks, often has a basin size of one in strongly

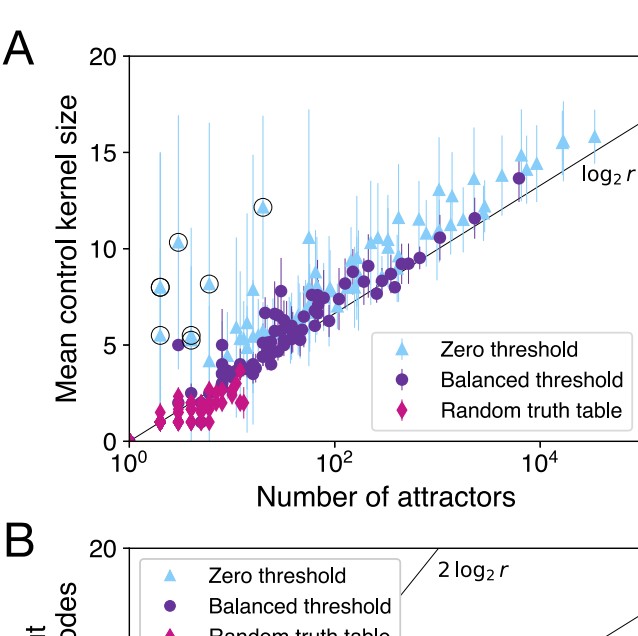

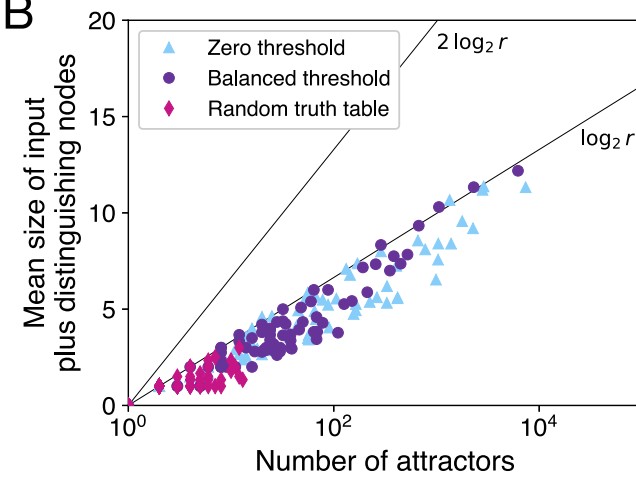

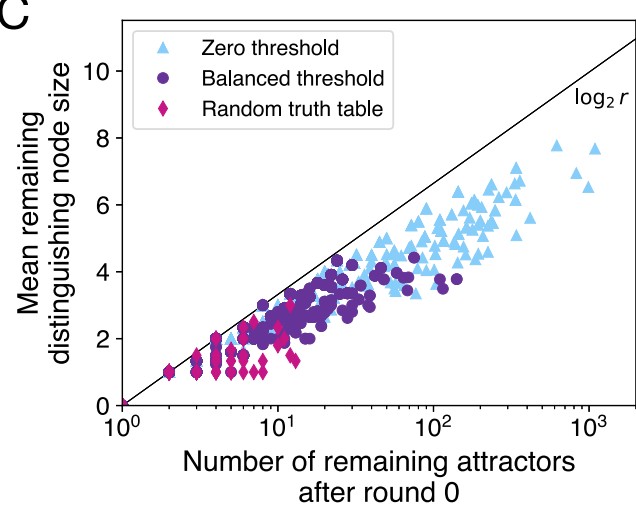

**Fig. 9 Control kernels in random networks show similar characteristics to those seen in the biological cases.** Here we choose networks of size 10, 15, and 20 from three ensembles: Erdös–Rényi networks with zero thresholds (azure triangles), Erdös–Rényi networks with balanced threshold inputs (dark purple circles), and p-K networks with random truth tables (pink diamonds). **A** Mean control kernel sizes are typically near the base-2 logarithm of the number of attractors. Compare to Fig. 1. Outliers, highlighted with a black circle, share a bias toward excitation that creates repellor fixed points. **B** The first-order approximation to control kernel size is typically below $\log_2 r$ and always below $2\log_2 r$. Compare to Fig. 8. **C** After fixing input nodes in round zero, distinguishing node sizes are empirically bound by $\log_2 r$. Compare to Fig. 7.

logarithmic scaling with respect to the number of attractors (dark purple circles in Fig. 9A).

We see in Fig. 9B, C that patterns in the size of the first-order approximation to the CK are similar to the biological networks: the number of input plus distinguishing nodes is typically near or below the base-2 logarithm of the number of attractors (and always below $2\log_2 r$; Eq. (5)), and the number of distinguishing nodes is always below the logarithm of the number of attractors they distinguish (Eq. (4)).

## Discussion

We analyzed control properties of a sizable fraction of the largest database of biologically inspired, Boolean networks currently available[30]. Following[6], we quantified the amount of external control necessary to drive a network to one of its original steady states in terms of the number of nodes whose states must be pinned. We found that the average amount of control necessary scales as the logarithm of the number of steady states of the unconstrained dynamics. At least within the range of network sizes we explored, this scaling is largely unaffected by the network size.

We show that the scaling is explained by the first two contributors to the CK: the number of input nodes plus the number of nodes whose value must be specified to distinguish the desired state from the other possible steady states. On average, these two contributions account for most of the CK in the biological cases we analyze (Fig. 4). We explored the limits of this result and exhibited two scenarios that can invalidate it. First, the projective plane provides a highly contrived example that violates the logarithmic scaling of distinguishing nodes. Second, there are examples for which convergence of the iterative procedure is not fast, including biased dynamics that create fixed points with extremely small basins. We leave as open questions whether the absence of these exceptions within the biological models might be due to their biological nature, to their tendency for having near-critical sensitivity[43], or to the fact that these networks are designed to be easily interpretable.

CKs are fundamentally related to closed circuits in the regulatory networks, as suggested by the close relation to the methods of stable motifs and feedback vertex sets that rely on identifying these closed circuits. In particular, we expect that each non-input CK node is part of a directed closed circuit in the regulatory networks (as paths in the network not involved in cycles correspond to deterministic cascades that cannot support multiple possible states).

We expect our results to have important biological implications. First note that, differently from[6], we did not restrict our analysis to selecting one particular steady state (the "main attractor" with largest basin). On the contrary, we considered all possible fates of the dynamics. In this expanded analysis, we showed that the size of CKs corresponding to different fates

biased networks. We might call such a fixed point an unstable "repellor" in the sense that all perturbations lead away from the state. Such a fixed point typically has a CK consisting of all or most nodes of the network, even when there are few overall attractors. All circled points in Fig. 9A have such a repellor fixed point (see "Outlier random networks" in Supplementary material). These repellor cases are a concrete example of how biases in the dynamics can lead to additional nodes making a non-negligible contribution to the CK. When the bias toward activation is removed in the "balanced" networks, these exceptions disappear and CK sizes are again characterized by

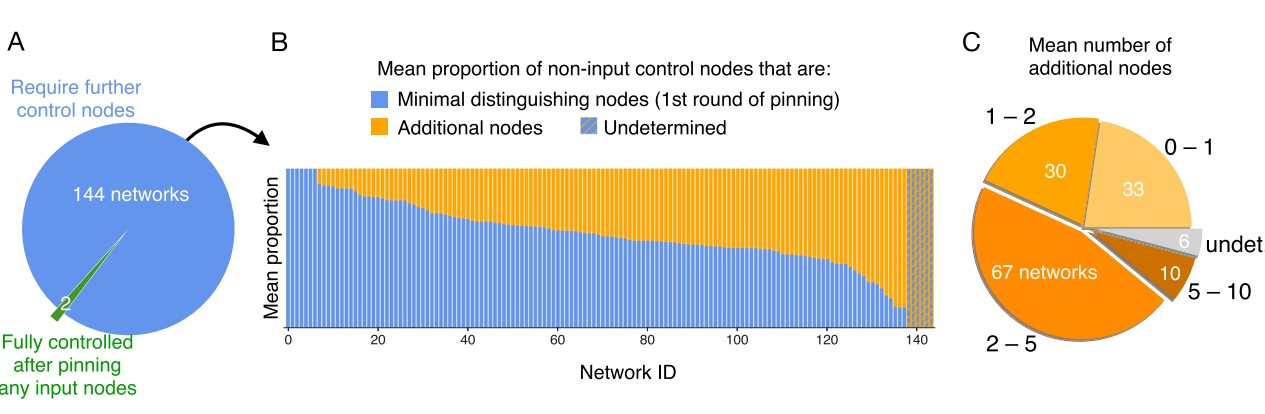

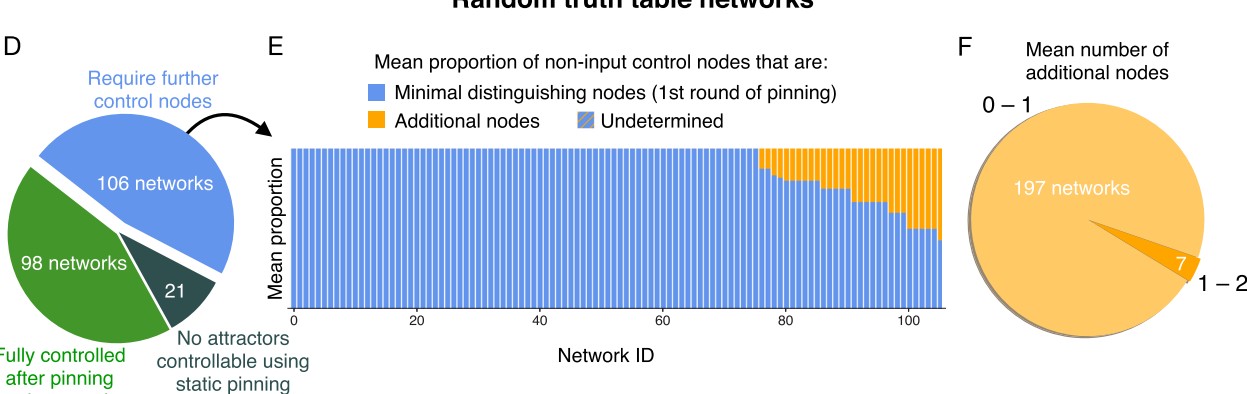

**Fig. 10 Contributions to control kernel size for random networks.** Compare to Fig. 4. For descriptions of random ensembles, see "Methods". **A–C** Control kernel contributions for random threshold networks (including biased and balanced cases). These networks often require additional control nodes, but the relative number of additional nodes is still often smaller or comparable to the first-order approximation given by the minimal distinguishing nodes. **D–F** Control kernel contributions for random truth table networks. These networks are typically easier to control. In **D**, we also see some networks for which all attractors are cycles and none are controllable using static pinning.

exhibit a remarkably small variance (error bars in Fig. 1). A small variance in CK size would suggest that converting among alternative cell types may be easy for any cell type, whereas large variance could be caused by, for example, adult cell types that are relatively easy to control for (small CK) but are prohibitively difficult to reprogram back to their progenitors (large CK).

Our result may also be relevant to the interpretation of the large amount of data attained through the recent advancements in single-cell RNA-seq techniques[44]. Unsupervised clustering based on transcriptome profiles has already helped identifying otherwise elusive cell types (e.g.,[45,46]). At the same time, cell types obtained through unsupervised clustering are sometimes difficult to relate to classification schemes based on morphology and function[47]. We have seen that the number of cell types (attractors) can be greatly affected by the presence of extracellular control, suggesting the possibility that single-cell sequencing experiments might also be exhibiting what the cell types would be in isolation, without external control factors. On the contrary, morphology and function might be identifying the smaller number of cell types determined by cell-to-cell signaling within a tissue.

The observed logarithmic scaling implies that biological networks may in fact be as easily controlled on average as would be expected if each pinning simply removed half of the remaining possible behaviors. A regulatory network with tens of thousands of genes could easily admit ~100 attractors and appear extremely

difficult to control. But logarithmic scaling of control implies that forcing the expression of few accurately selected genes might be enough to reduce hundreds of cell types to just one type with recognizable morphology or function.

The two cases that break logarithmic scaling highlight possible strategies for designing systems that are more difficult to control. First, the projective plane example implies average CK sizes that grow as the square root of $r$, though this would require much larger networks with many more attractors than are typically analyzed, and it is likely to require highly constructed, non-random structure. Significant differences show up only in particularly large examples: for instance, selecting one out of a million states could require controlling roughly 1000 nodes in a projective plane example instead of 20 assuming logarithmic scaling. On the other hand, biasing dynamics to create very small basin sizes could be a more natural way to force CKs to be large, as in the biased threshold networks. Biological selection for systems that are difficult to control is an intriguing possibility.

More generally, in cases in which our assumptions hold, the rule of thumb that pinning a binary variable reduces the number of outcomes by a factor of two becomes more than just a vague intuition. In these cases, we are justified in focusing on distinguishing nodes as a first approximation of necessary control nodes. This greatly simplifies the search because distinguishing nodes depend only on the alternative long-term dynamics of the system, and not specifics about how the system gets there.

## Methods

**Upper bound on the first-order approximation of CK size**. Given a network with $r$ attractors and $m$ input nodes, we can split the attractors into $2^m$ disjoint sets, one for each configuration of the inputs. If $r_j$ is the number of attractors sharing the $j$th input configuration, $\sum_{j=1}^{2^m} r_j = r$. Therefore,

$$\langle |w^{(1)}| \rangle = \frac{1}{r}\sum_{i=1}^{r} |w_i^{(1)}| = \frac{1}{r}\sum_{j=1}^{2^m} r_j \left( \frac{1}{r_j}\sum_{k=1}^{r_j} |w_k^{(1)}| \right).$$

We know the inequality $\frac{1}{r_j}\sum_{k=1}^{r_j} |w_k^{(1)}| \leq \log_2 r_j$ is not exact, as the projective plane construction represents a known exception. Nonetheless, it is exact within the networks we analyze (see Fig. 7). We want to show that assuming its validity leads to an upper bound on $\langle |w^{(1)}| \rangle$:

$$\langle |w^{(1)}| \rangle \leq \frac{1}{r}\sum_{j=1}^{2^m} r_j \log_2 r_j = \sum_{j=1}^{2^m} \frac{r_j}{r}\log_2 \frac{r_j}{r} + \log_2 r. \quad (7)$$

Let us call $d = 2^m$ and $\mathbf{x} = (r_1, ..., r_d)/r$. The first term in (7), $f(\mathbf{x}) = \mathbf{x} \cdot \log_2 \mathbf{x}$ is a convex function, and we can find its extreme values as usual. We first extend it to real values, then find its minimum value outside the probability simplex $\Delta^{d-1}$, i.e., the domain with $\mathbf{x} \cdot \mathbf{1} = 1$ and $x_j \geq 0, \forall j$. As $\nabla f(\mathbf{x}) = (\ln \mathbf{x} + \mathbf{1})/\ln 2$, $f$ has a minimum point at $\bar{\mathbf{x}} = 1/e$. The line identified by $\bar{\mathbf{x}}$ is already orthogonal to the simplex. Therefore, its projection onto $\Delta^{d-1}$ is simply $\mathbf{1}/d$. By virtue of the convexity of $f$, its actual minimum will be reached by the integers $r_j$ that best match an even distribution of the attractors among the $2^m$ sets. (This is expected, as $f$ is just a negative Shannon entropy.) When this happens,

$$\sum_{j=1}^{2^m} \frac{r_j}{r}\log_2 \frac{r_j}{r} \gtrsim -m.$$

The maximum of $f$ is reached near to each vertex of the simplex, when one $r_j$ is at its maximum (i.e., $1/r$), while all others are at their minimum (i.e., $1 - (2^m - 1)/r$). This gives the following upper bound on $\langle |w^{(1)}| \rangle$:

$$\langle |w^{(1)}| \rangle \leq \left(1 - \frac{2^m - 1}{r}\right)\log_2\left(1 - \frac{2^m - 1}{r}\right) + \left(1 - \frac{2^m - 1}{r}\right)\log_2 r. \quad (8)$$

For $m = 0$, we regain our hypothesis applied to the entire set of attractors, $\langle |w^{(1)}| \rangle \leq \log_2 r$. For $m = \log_2 r$, $\langle |w^{(1)}| \rangle = 0$.

Further, by noting that both $m$ and $\langle |w^{(1)}| \rangle$ are bounded by $\log_2 r$ when inequality (4) holds, we can place a simple, conservative upper bound on the "first-order approximation" of $|CK|$ that depends only on $r$:

$$\langle |CK^{(1)}| \rangle = m + \langle |w^{(1)}| \rangle \leq 2\log_2 r.$$

We find empirically that the actual $\langle |CK^{(1)}| \rangle$ values lie significantly below this conservative bound. For example, in the biological cases we test, $\langle |CK^{(1)}| \rangle$ never exceeds $\log_2 r$ (see Fig. 8). The reason for this can be understood by analyzing Eq. (7) more closely. Equation (7) originates from the assumption that $\frac{1}{r_j}\sum_{k=1}^{r_j} |w_k^{(1)}| \leq \log_2 r_j, \forall j$. Smaller values of $|w_k^{(1)}|$ for individual attractors can cause $\langle |w^{(1)}| \rangle$ to lie below the maximum given by Eq. (7). But this maximum itself varies depending on the distribution of the number of attractors across input sets $j$, captured by the first term in Eq. (7), $f(\mathbf{x})$. We have seen that this term is a negative entropy, and that it approaches $-m$ when the $r$ attractors are (approximately) evenly distributed among the $2^m$ input configurations, that is to say when $r_j \simeq r/2^m$. By defining the input entropy $\mu = -f(\mathbf{x})$, we can express the upper bound on $\langle |CK^{(1)}| \rangle$ as:

$$\langle |CK^{(1)}| \rangle \leq \log_2 r + (m - \mu). \quad (9)$$

For evenly distributed attractors, $\mu \simeq m$, and $\langle |CK^{(1)}| \rangle \lesssim \log_2 r$. At the opposite extreme, if attractors are maximally unevenly distributed, $\mu$ can approach zero.

In the biological networks we test, we empirically find that $m - \mu$ is always too small to force $\langle |CK^{(1)}| \rangle$ above $\log_2 r$ (that is, $m - \mu < \log_2 r - \sum_j \frac{1}{r_j}\sum_{k=1}^{r_j} |w_k^{(1)}|$). Figure S6 in the Supplementary material shows that $\mu$ is always close to $m$ in these cases.

With the larger statistics provided by the ensembles of random networks, we find a few cases in which $\langle |CK^{(1)}| \rangle$ is slightly greater than $\log_2 r$ (four cases in the zero threshold ensemble and four more cases in the balanced threshold ensemble; see Fig. 7B), but still never greater than $2\log_2 r$, as expected.

**Modular approach to finding attractors**. Biological regulatory network models are in many cases highly modular (having subsets of nodes that interact much more with one another than with other subsets) and hierarchical (having upstream modules that do not depend on the behavior of downstream modules). As recognized before in the literature[48], the attractors of a Boolean network can be computed much more efficiently in modular hierarchical networks by analyzing modules separately. The basic idea is to find all attractors of upstream modules first, then use each upstream attractor to find corresponding attractors of downstream modules. As we detail in the following paragraphs, some subtleties make this process complicated, but it is still straightforward to accomplish computationally.

We begin with a decomposition of a given Boolean dynamical system into hierarchical modules, which we define here as strongly connected components of the system's causal network (with individual Boolean variables represented as nodes and dependencies between variables represented as directed edges). This produces a directed acyclic graph describing the dependencies among modules. First, for upstream modules that do not depend on any other modules, we find attractors using a brute-force approach that iterates over all possible states of the module. Then, for each downstream module $\mathcal{D}$, we iterate over all possible combinations of attractors $\mathbf{U}$ of upstream modules on which $\mathcal{D}$ depends. For a given attractor $U \in \mathbf{U}$, the states of nodes in the upstream modules are fixed to the values (perhaps dynamically varying, in the case of a cyclic attractor) that they have in $U$. Corresponding attractors for $\mathcal{D}$ are then found through a brute-force approach iterating over all possible dynamics of $\mathcal{D}$ given the input provided by $U$.

Two subtleties arise when computing the set of all possible attractors $\mathbf{U}$ for the nodes upstream of $\mathcal{D}$. In the easiest case, $\mathcal{D}$ depends on a number of separate modules $\mathcal{U}_1, \mathcal{U}_2, ..., \mathcal{U}_n$ that do not share ancestor modules further upstream, and all corresponding sets of upstream attractors $\mathbf{U}_1, \mathbf{U}_2, ..., \mathbf{U}_n$ consist only of fixed points. In this case, $\mathbf{U}$ consists of $\prod_i |\mathbf{U}_i|$ attractors, all combinations in which one attractor is chosen from each set of upstream attractors $\mathbf{U}_i$. The first subtlety arises when upstream modules share ancestor modules further upstream (for instance, if $\mathcal{U}_1$ and $\mathcal{U}_2$ both depend on $\mathcal{U}_\alpha$). In this case, some combinations of attractors will be inconsistent in that they do not have the same values on overlapping nodes in ancestor modules; these inconsistent combinations should be simply removed to not appear in $\mathbf{U}$. The second subtlety occurs with cyclic attractors, those with length $\ell > 1$. When combining two cyclic attractors $U_1$ and $U_2$ of upstream modules, with corresponding lengths $\ell_1$ and $\ell_2$, we must account for all possible phase shifts between the two attractors. This leads to $p$ distinct possible attractors, each of length $\ell_c$, where $p$ is the greatest common divisor of $\ell_1$ and $\ell_2$, and $\ell_c$ is their least common multiple.

**Computing control kernels**. Given our definition of CKs, a brute-force method for computing them is straightforward. For each attractor, we loop over sets of distinguishing nodes of increasing size. Pinning the distinguishing nodes to their values in the attractor, a CK is found when no other attractors exist in the pinned dynamics. Cyclic attractors that are not controllable are easily identified by pinning all constant nodes: if more than one attractor remains in this case, the cycle does not have a CK.

The modular approach to analyzing Boolean dynamical systems can also make the computation of CKs much more efficient. Note that, within a given module, CK nodes can be determined without analyzing downstream modules because any control exerted downstream of the module will not affect it. For this reason, in the modular approach we compute CK nodes at the same time we compute attractors for each module. That is, given each attractor state $U$ of upstream modules and for each attractor of the current module $\mathcal{D}$, we compute CK nodes within $\mathcal{D}$ by looping over distinguishing node sets of increasing size (restricted to nodes within $\mathcal{D}$) and pinning them until the dynamics produce a single attractor.

In some cases it is possible to compute CKs using the modular approach but computationally infeasible to compute minimal distinguishing node sets (because this requires checking every possible set smaller than the minimal size). In these cases we compute an upper bound on the size of the minimal distinguishing node sets (shown as triangles in Figs. 7 and 8), equal to at worst the size of the CK, and in some cases set by checking a limited subset of smaller distinguishing node sets that have been identified for other attractors in that network.

**Sampling analysis**. Some networks in the database contain modules that are too large to be analyzed exactly using the above approach. In particular, when a module has more than about 30 nodes, our code is unable to run the analysis using a reasonable amount of time or memory. In these cases, we also attempt a sampling analysis in which we find attractors by initializing the system at $N_S = 10^4$ to $10^6$ random states. We then use the brute-force method for finding CKs described above, where a CK is defined as producing the single desired attractor when initializing the system using the same $N_S$ random states. We first restrict sampling analysis to networks for which we find fewer than $10^3$ attractors.

This sampling analysis initially allowed us to find CK and distinguishing node data for seven additional networks. We then tested the dependence of our results on $N_S$ for these seven networks. For three of the networks, we confirmed the previous result when $N_S$ was increased by a factor of 10 to $10^5$. We achieved convergence for two of the remaining networks using a further factor of 10, finding the same attractors using $N_S = 10^6$. The remaining two networks continued to show non-negligible increase in the number of attractors, particularly for cyclic attractors with small basins. Most of these cycles were not controllable using a static intervention. We removed these two networks from our analysis, which did not significantly alter our results.

The five networks analyzed using only the sampling method are represented as black-bordered circles in Fig. 1 and unfilled circles in Fig. 7.

In Table 1, we list the names of the 49 networks we analyze, those for which we are able to find CKs for all attractors. Those names marked with an asterisk were analyzed using the sampling approach.

## Table 1 The 49 analyzed biological regulatory networks.

| | Network name | Size |
|---|---|---|
| 1 | Cortical Area Development | 5 |
| 2 | Cell Cycle Transcription By Coupled CDK And Network Oscillators | 9 |
| 3 | Mammalian Cell Cycle 2006 | 10 |
| 4 | Toll Pathway Of *Drosophila* Signaling Pathway | 11 |
| 5 | Metabolic Interactions In The Gut Microbiome | 12 |
| 6 | Regulation Of The L-arabinose Operon Of *Escherichia Coli* | 13 |
| 7 | Lac Operon | 13 |
| 8 | *Arabidopsis Thaliana* Cell Cycle | 14 |
| 9 | Cardiac Development | 15 |
| 10 | Predicting Variabilities In Cardiac Gene | 15 |
| 11 | Fanconi Anemia And Checkpoint Recovery | 15 |
| 12 | HCC1954 Breast Cell Line Short-term ErbB Network | 16 |
| 13 | Neurotransmitter Signaling Pathway | 16 |
| 14 | SKBR3 Breast Cell Line Short-term ErbB Network | 16 |
| 15 | BT474 Breast Cell Line Short-term ErbB Network | 16 |
| 16 | Body Segmentation In *Drosophila* 2013 | 17 |
| 17 | Budding Yeast Cell Cycle 2009 | 18 |
| 18 | T-LGL Survival Network 2011 Reduced Network | 18 |
| 19 | Vegf Pathway Of *Drosophila* Signaling Pathway | 18 |
| 20 | CD4+ T-Cell Differentiation And Plasticity | 18 |
| 21 | Oxidative Stress Pathway | 19 |
| 22 | Human Gonadal Sex Determination | 19 |
| 23 | Budding Yeast Cell Cycle | 20 |
| 24 | Mammalian Cell Cycle | 20 |
| 25 | Iron Acquisition And Oxidative Stress Response In *Aspergillus Fumigatus*. | 22 |
| 26 | B Cell Differentiation | 22 |
| 27 | T-Cell Differentiation | 23 |
| 28 | FGF Pathway Of *Drosophila* Signaling Pathways | 23 |
| 29 | HH Pathway Of *Drosophila* Signaling Pathways | 24 |
| 30 | Processing Of Spz Network From The *Drosophila* Signaling Pathway | 24 |
| 31 | TOL Regulatory Network | 24 |
| 32 | SKBR3 Breast Cell Line Long-term ErbB Network | 25 |
| 33 | HCC1954 Breast Cell Line Long-term ErbB Network | 25 |
| 34 | BT474 Breast Cell Line Long-term ErbB Network | 25 |
| 35 | *Trichostrongylus Retortaeformis* | 26 |
| 36 | Pro-inflammatory Tumor Microenvironment In Acute Lymphoblastic Leukemia | 26 |
| 37 | Wg Pathway Of *Drosophila* Signaling Pathways | 26 |
| 38 | Death Receptor Signaling | 28 |
| 39 | FA BRCA Pathway* | 28 |
| 40 | Septation Initiation Network | 31 |
| 41 | Tumor Cell Invasion And Migration | 32 |
| 42 | *Bordetella Bronchiseptica** | 33 |
| 43 | Lymphoid And Myeloid Cell Specification And Transdifferentiation* | 33 |
| 44 | Cholesterol Regulatory Pathway* | 34 |
| 45 | T-Cell Signaling 2006 | 40 |
| 46 | Treatment Of Castration-Resistant Prostate Cancer | 42 |
| 47 | Guard Cell Abscisic Acid Signaling | 44 |
| 48 | Pc12 Cell Differentiation* | 62 |
| 49 | Yeast Apoptosis | 73 |

**Iterative bound on control kernel size**. We used the sampling method to run the iterative algorithm that computes the upper bound in Eq. (2) to make Fig. 5. This produces iterative bound data for 40 networks. When compared to the true average CK size, the iterative upper bound is often tight (see Fig. S7 in the Supplementary material).

**Random network ensembles**. We use two types of random network ensembles, one in which node logic is defined in terms of truth tables and one in which logic is determined by thresholds to activation.

To create random truth tables, we construct $p$–$K$ networks in which each node depends on $K = 2$ other randomly chosen nodes. Each node's truth table is constructed such that each of the $2^K = 4$ possible states of its input nodes has a

probability $p$ of activating the node. We sampled 225 networks from this ensemble, with 75 each having $p = 0.25$, 0.5, and 0.75, and 25 each within these sets having number of nodes $n = 10$, 15, and 20. We were successful in finding CKs for all controllable attractors for all of these sampled networks.

To create random threshold networks, we follow ref. [6]. The network's dependency structure $A$ is first chosen as an Erdös–Rényi graph with average degree $d$, and each edge in this graph is assigned with probability $p_I$ a value of $-1$ (representing an inhibitory interaction), or otherwise $+1$ (representing an excitatory interaction). Each node's state is determined by comparing the sum of the incoming signed inputs $s_i = \sum_j A_{ij} x_j(t)$ to its threshold $\tau_i$:

$$x_i(t+1) = \begin{cases} 0, & \text{if } s_i < \tau_i \\ x_i(t), & \text{if } s_i = \tau_i \\ 1, & \text{if } s_i > \tau_i. \end{cases} \quad (10)$$

We consider two choices for the thresholds $\tau$. In the first case, all thresholds are set to $\tau = 0$; in the second "balanced" case, thresholds are set to half of the sum of incoming edges: $\tau_i = \frac{1}{2}\sum_j A_{ij}$. We sampled 75 networks from this ensemble using each type of threshold, with each combination of $d = \{1, 2, 3, 4, 5\}$, $p_I = \{0.1, 0.3, 0.5, 0.7, 0.9\}$, and $n = \{10, 15, 20\}$, for a total of 150 networks. Of these, for 146 we were successful in finding CKs for all controllable attractors.

**Reporting summary**. Further information on research design is available in the Nature Research Reporting Summary linked to this article.

## Data availability
The control kernel size data generated in this study have been deposited in as a Zenodo repository under the accession code https://doi.org/10.5281/zenodo.5172898. The biological network models analyzed in this study are available on the Cell Collective website (https://cellcollective.org/).

## Code availability
The python code used to calculate control kernels is available as a Zenodo repository under the accession code https://doi.org/10.5281/zenodo.5172898. This code depends on a number of open-source software packages: neet, datadex, numpy, and networkx.

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

## Acknowledgements

We thank Cyrus Rashtchian for identifying the number of distinguishing nodes as a problem of witness sets in computational learning theory, and we thank Cole Mathis for useful comments on an early draft. We thank Doug Moore for contributions to the modular analysis code. We acknowledge Research Computing at Arizona State University for providing computing resources. B.C.D. was supported by a fellowship at the Wissenschaftskolleg zu Berlin.

## Author contributions

E.B. and B.C.D. designed the research, analyzed data, and wrote the paper.

## Competing interests

The authors declare no competing interests.
