## [Peer Review File · Nature Communications]

The basis of easy controllability in Boolean networksReviewers' Comments:

Reviewer #1:

Remarks to the Author:

The authors showed that the size of CK depends on the number of attractors irrespective of the network size and that it is bounded by base-2 logarithm scaling. To probe its reason, they suggested an iterative procedure to estimate CK and classified CK into three types. The result is interesting, but this reviewer wonders the validity of assumptions and supporting results. And, the relationship between the size of CK and the number of attractors was also investigated in the original paper (Kim, J., Park, S.-M. & Cho, K.-H. Discovery of a kernel for controlling biomolecular regulatory networks. *Scientific Reports* 3, 2223 (2013)), so the authors need to more clearly describe the novelty of their work.

Major comments

1. The authors assumed particular dynamics (p. 10, lines 250-261) and mentioned that the expectation value of the number of fixed points is one when there is no input. However, real biological networks can have multiple fixed points due to positive feedbacks even if there is no input (see Kwon, Y.-K. & Cho, K.-H. Boolean Dynamics of Biological Networks with Multiple Coupled Feedback Loops. *Biophys J* 92, 2975-2981 (2007)). Actually, Figure 4 indicates that the mean of attractor numbers is not close to one and that this might be because the biological networks have different dynamics from the assumption of this study. This reviewer is surprised to see that the size of CK still shows logarithmic scaling with respect to the number of attractors in spite of such discrepancy of assumptions. This raises a question whether all the remaining networks do not have any positive feedback during the iterative procedure.

2. This reviewer suspects that the distinguishing node might be related to feedback vertex set or differentially expressed positive circuits (Mochizuki, A., Fiedler, B., Kurosawa, G. & Saito, D. Dynamics and control at feedback vertex sets. II_ A faithful monitor to determine the diversity of molecular activities in regulatory networks. *J Theor Biol* 335, 130-146 (2013); Crespo, I., Perumal, T. M., Jurkowski, W. & Sol, A. del. Detecting cellular reprogramming determinants by differential stability analysis of gene regulatory networks. *Bmc Syst Biol* 7, 140 (2013).), So, the authors need to compare it with them.

3. Figure 4 only shows the number of attractors remained after pinning, but 262-267 lines on p. 10 mention the number of additional nodes which needs to be further shown in the figure such that we can verify Eq. 3.

4. The authors mentioned that cycle is not a dominant feature, which seems to be not supported if we see Figure 5. Many networks have such cyclic attractors (even one third of 51 networks have more than half of their attractors as cyclic ones). The authors need to clarify this. They may show that the proportion of cyclic attractors is small for those attractors of large basins.

5. The authors used 10^4 initial states to estimate attractors and their CK when they cannot exactly find the attractors. This reviewer wonders whether 10^4 is a sufficient number compared to the whole state space. So, the authors need to verify their results by increasing the number of initial states.

6. The authors considered only the networks in Cell Collective, but they need to further analyze random networks and see whether the their claim holds for general networks.

7. The following descriptions are unclear:

A. p.4 line 106

Definition 1 (Control Kernel): For a given attractor A, a control kernel CK_j is defined ... what "CK_j" means?

B. p.4 line 113

whether all starting points converge to A... what "points" mean?

C. p.7 line 179

Note that it is possible for nodes that were not input nodes in the initial network to become effective input nodes after this pinning procedure. A simple example is a node whose state depends only on input nodes... does "effective input nodes" be further pinned in round 0 or in the first round as distinguishing node?

D. The iterative pinning procedure in Fig. 2 on p. 10 and the computing control kernels on p. 19 are confusing. Is it correct that the iterative pinning procedure is to estimate the bound for CK and the actual CK is to be found by using the method of p. 19?

E. What is "an ensemble of dynamics" on line 250, p. 10?

F. It is very difficult to understand the "Finite Projective Plane" on p. 16 without seeing all the details in Suppl., so the authors need to further elaborate it (by moving some contents in Suppl. to main manuscript).

Minor comments

1. Figure 3, y-axis needs in the right bar graph.
2. p.10 line 256, "right size" should be "right side".
3. Figure 7, $w(0)$ should be $w(1)$?
4. The upper bound of the first order approximation, $2\log_2(r)$ needs to be indicated in Figure 7.

Reviewer #2:

Remarks to the Author:

This manuscript analyzes the attractor (phenotype) controllability of 51 Boolean models of biological networks using a previously developed control kernel method. The analysis yields that the average number of nodes that must be controlled to drive the system into one of its natural attractors scales logarithmically with the number of original attractors. This is a mild increase, which is good news for practical applications. Unfortunately, the analysis is not as thorough as it should be and important comparisons are missing. I will elaborate below.

1. Early in the introduction the manuscript states "Common approaches to finding interventions to steer a complex system from an undesirable to a desirable state currently rely on heuristic genetic and greedy algorithms to bound the size of minimal interventions." This is not a correct assessment. In fact, there is powerful and general theory for attractor control that gives a proven upper bound for minimal interventions that can drive a dynamical system into a desirable attractor. This theory was published in two foundational papers in 2013:

Fiedler et al., Dynamics and Control at Feedback Vertex Sets. I: Informative and Determining Nodes in Regulatory Networks, *Journal of Dynamics and Differential Equations*, volume 25, pages 563–604 (2013)

Atsushi Mochizuki, Bernold Fiedler, Gen Kurosawa, Daisuke Saito, Dynamics and control at feedback vertex sets. II: A faithful monitor to determine the diversity of molecular activities in regulatory networks, *Journal of Theoretical Biology*, Volume 335, Pages 130-146 (2013)

The original publications focused on networks that do not have source (input) nodes. The theory was

extended to such networks in

Jorge G T Zanudo, G Yang, R Albert, Structure- based control of complex networks with nonlinear dynamics. *Proceedings of the National Academy of Sciences of the United States of America*, 114(28) (2017). This article also describes attractor control of Boolean systems and demonstrates that for any specific model the number of control nodes is smaller than the upper bound given by Feedback Vertex Set theory. This paper is cited as reference 4 in the manuscript. It is not clear why the authors, having read this article, chose not to disclose that an upper limit to the control nodes already exists: it is made up by all source nodes plus the feedback vertex set of the network.

2. The manuscript uses the control kernel method introduced by Kim et al in 2013. This is one of multiple methods available for attractor control of Boolean networks. For example, another method is based on control of stable motifs, as described in Jorge G T Zanudo & R Albert, Cell fate reprogramming by control of intracellular network dynamics, *PLOS Computational Biology* 11(4): e1004193 (2015). The manuscript needs to support the results by employing multiple methods.

3. The identification of the number of attractors is a key component of the work. The authors seem not to be aware of the fact that a multitude of methods exist to identify attractors in Boolean networks. As an example, the article describing the CoLoMoTo Interactive Notebook (Naldi et al. *The CoLoMoTo Interactive Notebook: Accessible and Reproducible Computational Analyses for Qualitative Biological Networks*, *Frontiers in Physiology* 2018) describes some of these methods and their implementation. The results in terms of the number of attractors need to be verified by another method.

4. Another point related to the number of attractors. It is well documented that a large subset of cyclic attractors of synchronous Boolean models are artifacts: they rely on the synchronicity of node state changes and are destroyed for any perturbations to that synchronicity. The number of attractors decreases dramatically in asynchronous networks (see for example Greil F, Drossel B. Dynamics of critical Kauffman networks under asynchronous stochastic update. *Phys Rev Lett.* 2005;95: 048701.). The number of synchronous attractors is not a very good reflection of the phenotype diversity of biological systems. Thus, the identified logarithmic scaling may be due to an inflation of the number of attractors.

REVIEWER COMMENTS

Reviewer #1

(Expertise: Boolean GRNs):

The authors showed that the size of CK depends on the number of attractors irrespective of the network size and that it is bounded by base-2 logarithm scaling. To probe its reason, they suggested an iterative procedure to estimate CK and classified CK into three types. The result is interesting, but this reviewer wonders the validity of assumptions and supporting results. And, the relationship between the size of CK and the number of attractors was also investigated in the original paper (Kim, J., Park, S.-M. & Cho, K.-H. Discovery of a kernel for controlling biomolecular regulatory networks. *Scientific Reports* 3, 2223 (2013)), so the authors need to more clearly describe the novelty of their work.

While the analysis in [Kim et al 2013] is focused on the primary attractor (the attractor with the largest basin), we investigate the controllability of all attractors of each network. The scaling property we find holds for the average size of such control kernels. The analysis of 8 biological models is discussed in Kim et al. 2013 (with one of these appearing also in our analysis). The authors then extended their technique to the analysis of several ensembles of random networks in their supplementary document. With larger statistics, their results exhibit a scaling behavior compatible with our findings (their Fig. S3), but no theoretical interpretation is provided, other than a heuristic explanation in terms of the number of inhibitory links. Generalizing the study to attractors other than the primary one allowed us to better understand the reasons for such scaling.

We now make clear the relation of our work with [Kim et al 2013] in the Introduction:

“Here, across 49 example biological networks, we compute control kernels for all attractors. We first corroborate earlier results that CKs remain

relatively small, finding typical CK sizes smaller than about a dozen nodes even in large networks with up to 72 interacting components.

More importantly, we illuminate the origins of this easy controllability by showing that the average CK sizes do in fact scale logarithmically with the number of attractors. For a number of reasons, this close correlation with the number of attractors has not been highlighted in prior studies. In particular, the scaling is not as clearly evident when looking only at the attractor most often reached from random starting points ... “

Major comments

1. The authors assumed particular dynamics (p. 10, lines 250-261) and mentioned that the expectation value of the number of fixed points is one when there is no input. However, real biological networks can have multiple fixed points due to positive feedbacks even if there is no input (see Kwon, Y.-K. & Cho, K.-H. Boolean Dynamics of Biological Networks with Multiple Coupled Feedback Loops. *Biophys J* 92, 2975–2981 (2007)). Actually, Figure 4 indicates that the mean of attractor numbers is not close to one and that this might be because the biological networks have different dynamics from the assumption of this study. This reviewer is surprised to see that the size of CK still shows logarithmic scaling with respect to the number of attractors in spite of such discrepancy of assumptions. This raises a question whether all the remaining networks do not have any positive feedback during the iterative procedure.

We thank the reviewer for pointing out a possible source of misunderstanding in the previous version of our manuscript. The discrepancy is not in the assumptions, but in what we meant by “order 1” in the previous version of the manuscript. In a specific class of random networks [Samuelsson & Troein 2003] the average number of fixed point attractors is 1, irrespective of the size of the network. These RBNs can have feedback loops, but no input nodes. In the biological networks we analyze, after iterative pinning steps, we find just one attractor more than half the time, and a number much greater than one in less than 10% of the cases. This is

what we meant by “order 1” in the previous version of our manuscript, and what is shown in the figure the reviewer refers to (the current Fig. 5, previously Fig. 4).

We stress that our argument does not rely on the average number of attractors being exactly one. Instead, in order for the pinning procedure to terminate quickly, the probability of having a single attractor after pinning needs to be large, and in order to add a small number of additional nodes at each step, the typical number of attractors needs to be small. To get a rough expectation for the number of attractors after pinning, we use a similar argument to [Samuelsson & Troein 2003], where the average number of fixed-point attractors can be found analytically for random networks. But we do that only to get additional insights for understanding networks that are outliers with respect to the scaling we uncover.

We have now reorganized and rephrased much of the section “First order approximation of the control kernel” in order to clarify this point. In particular, we have removed our previous vague language of “order 1”, more clearly stated that the expectation of having one fixed point attractor applies only to RBNs, and added text that more clearly states the assumptions necessary to produce logarithmic scaling (corresponding to our approximation that neglects “additional” nodes):

“With a small number of attractors remaining after each round of pinning, there is a fairly high probability of having, at some step, only one attractor, at which point a CK has been found and iteration stops. This probability, which we empirically observe to be 58% across the networks we study (Fig. 5), confirms our simple interpretation.

We have anticipated that our first order approximation of the $|CK|$ consists of neglecting the additional nodes (the lower bound in Eq. 2). A fast convergence of the pinning procedure guarantees a small number of rounds of pinning (the number of nonzero terms $|w^{(i)}|$ with $i \geq 2$ in Eq. 2) ...

...

[W]e have reason to expect that witness set sizes, and therefore distinguishing node sets, are typically no larger than the logarithm of the number of attractors they distinguish. In each round of pinning after the first, the expectation that we have only a few remaining attractors then translates into small numbers of additional nodes added in each round (small terms $|w^{(i)}|$ in Eq. 2 when $i \geq 2$). This, combined with the expectation that only a few rounds contribute, leads us to consider an approximation in which we neglect the additional nodes.”

Finally, in response to the reviewer’s comments about positive feedback: Our results are in fact consistent with [Know & Cho 2007], who find an average number of fixed-points always less than 3 in small random Boolean networks. They find that positive feedback loops are associated with a larger ratio of fixed point attractors to cyclic attractors, but the absolute number of attractors, most important for our analysis, never becomes large.

2. This reviewer suspects that the distinguishing node might be related to feedback vertex set or differentially expressed positive circuits (Mochizuki, A., Fiedler, B., Kurosawa, G. & Saito, D. Dynamics and control at feedback vertex sets. II A faithful monitor to determine the diversity of molecular activities in regulatory networks. J Theor Biol 335, 130–146 (2013); Crespo, I., Perumal, T. M., Jurkowski, W. & Sol, A. del. Detecting cellular reprogramming determinants by differential stability analysis of gene regulatory networks. BMC Syst Biol 7, 140 (2013).), So, the authors need to compare it with them.

As also pointed out by Reviewer 2, **feedback vertex sets** are indeed closely related in that they provide an upper bound on the size of our control kernels. Specifically, a control kernel for any particular attractor cannot be larger than the smallest vertex set. Feedback vertex sets are nice in that they are efficient to compute (because they only take into account the network dependency structure, not the full dynamics at each node), but they can be considerably larger than the

attractor-specific control kernels we compute. To better connect with existing methods, we have now added a new subsection titled “Comparing our results to those obtained using alternative methods”. In the new Fig. 2D, we compute the size of feedback vertex sets for the networks in our database. Importantly, feedback vertex sets do not compare directly to the average size of our control kernels. Therefore, we compare their sizes to a closely analogous quantity in our analysis: the size of the union control kernel, consisting of all nodes that appear in attractor-specific control kernels across all attractors for each network. Interestingly, once plotted versus the number of attractors, feedback vertex sets also show a mild logarithmic scaling. The feature is less evident in that case, as we expect, because our result holds only with respect to the average size of the control kernels, as opposed to the union of such sets.

Additionally, **differentially expressed positive circuits (DEPC)** have been used to find nodes that, when forced, move the system from one of the network’s attractors to another of its attractors. Note that the goal of our control kernel is different: we want to get to the desired attractor from all starting points, whereas the DEPC approach explored in Crespo et al. only guarantees that the desired attractor will be reached when starting from a given initial attractor. We now mention DEPC when comparing our control kernel definition with other approaches in the Introduction.

3. Figure 4 only shows the number of attractors remained after pinning, but 262-267 lines on p. 10 mention the number of additional nodes which needs to be further shown in the figure such that we can verify Eq. 3.

We now include a new chart in Fig. 4C that directly shows that the mean number of additional nodes is typically small.

0. The authors mentioned that cycle is not a dominant feature, which seems to be not supported if we see Figure 5. Many networks have such cyclic attractors (even one third of 51 networks have more than half of their attractors as cyclic ones).

The authors need to clarify this. They may show that the proportion of cyclic attractors is small for those attractors of large basins.

We agree with the reviewer about the lack of clarity in our argument regarding the importance of cyclic attractors in the previous version of our manuscript. On further reflection, the number of initial cycles is not crucial to our argument. What really matters in our analysis is that the number of cycles does not proliferate while we perform our iterative pinning procedure. That could have been the case, as we know that large numbers of cycles can appear in RBNs of larger sizes than the ones we consider. Our current Fig. 5 (previously Fig. 4), showing the total number of attractors after pinning (including cycles), confirms that that does not happen in the biological networks we study. We have significantly revised the “Cyclic attractors” subsection in the “First order approximation of the control kernel” section to clarify this point, saying:

“A proliferation of cycles created during the intermediate rounds of pinning could potentially prevent our iterative procedure from converging quickly. Nonetheless, this is not something we observe in the biological networks we analyzed. Fig. 5 shows that the number of attractors remaining after each iterative pinning, including cycles, is typically small.”

We now also show our results for the 24 biological networks that possess only fixed-point attractors and compare the results to the 25 networks that do contain cycles. This result is shown in our new Fig. 13E and F, and confirms that cycles, when controllable, do not alter our main conclusion.

5. The authors used 10^4 initial states to estimate attractors and their CK when they cannot exactly find the attractors. This reviewer wonders whether 10^4 is a sufficient number compared to the whole state space. So, the authors need to verify their results by increasing the number of initial states.

We agree with the reviewer, and have progressively increased the number of sampling points N_s in the analysis of the seven networks for which sampling was needed. As we now state in the Methods:

“For three of the networks, we confirmed the previous result when N_S was increased by a factor of 10 to 10^5 . We achieved convergence for two of the remaining networks using a further factor of 10, finding the same attractors using $N_S = 10^6$. The remaining two networks continued to show non-negligible increase in the number of attractors, particularly for cyclic attractors with small basins. Most of these cycles were not controllable using a static intervention. We removed these two networks from our analysis, which did not significantly alter our results.”

6. The authors considered only the networks in Cell Collective, but they need to further analyze random networks and see whether their claim holds for general networks.

We agree with the reviewer, and we have now complemented our manuscript with the analysis of three ensembles of random networks. As described in the new section “Random networks” and the new Figures 9 and 10, the logarithmic scaling remains clear in the random networks, with a few exceptional cases that demonstrate how biases in the dynamics can lead to a larger number of additional control nodes being needed. (We note that our claim was never that *all* networks follow logarithmic scaling of CK size with number of attractors, but rather that many do, and that we can characterize a number of exceptions to this apparent rule.)

We first consider random threshold networks as previously considered in the SI of Kim *et al.* 2013. The first ensemble (azure triangles in the new Fig. 9) assumes zero thresholds for all nodes, exactly as in the cited work. We expect this ensemble to be biased toward attractors that have more activated than inactivated nodes, because nodes are very easily activated. In particular, we find this behavior in all eight outlier networks highlighted in Fig. 9A as circled triangles (with additional

details in the Supplementary Material section “Outlier random networks”): the control kernel for the all-inactive state consists of all n nodes because they must all be pinned to an inactive state in order for activation not to spread through the system. These are examples in which the controlled dynamics does not converge quickly under pinning (many rounds of pinning are needed).

To investigate whether a less biased ensemble could remove these outliers, we also consider an ensemble of “balanced” threshold networks (purple circles in the same figure). In this ensemble, each node’s threshold is set such that only half the total configurations of incoming signals will activate it. This less biased scenario shows, as expected, far fewer exceptions from our predicted scaling.

Finally, we consider p - K random networks with fixed in-degree and randomized truth tables, which typically have fewer attractors and follow our expectations in having small CK sizes.

7. The following descriptions are unclear:

A. p.4 line 106

Definition 1 (Control Kernel): For a given attractor A , a control kernel CK_j is defined ... what “ CK_j ” means?

The subscript j was referring to the different possible CKs of the same attractor. That notation has been simplified in the revised manuscript.

B. p.4 line 113

whether all starting points converge to A ... what “points” mean?

We meant the initial states of the network. The text has been updated with this more precise terminology.

C. p.7 line 179

Note that it is possible for nodes that were not input nodes in the initial network to become effective input nodes after this pinning procedure. A simple example is a node whose state depends only on input nodes... does “effective input nodes” be further pinned in round 0 or in the first round as distinguishing node?

We do not make this distinction in the code. Effective input nodes would be counted among the distinguishing nodes. We have added a footnote to the manuscript to avoid this source of confusion.

D. The iterative pinning procedure in Fig. 2 on p. 10 and the computing control kernels on p. 19 are confusing. Is it correct that the iterative pinning procedure is to estimate the bound for CK and the actual CK is to be found by using the method of p. 19?

Yes, this interpretation is correct, and this was not as clear as it could be in the original text. While we use the iterative process to construct bounds on the control kernel size (as in Eq. 2), our results in Fig. 1 for the actual size of control kernels are computed using a different algorithm as described in the methods section. We now make this clear in the caption for Fig. 3 (corresponding to the original Fig. 2).

E. What is “an ensemble of dynamics” on line 250, p. 10?

That section has changed significantly during our review. We meant the average properties of an ensemble of networks. We believe the new text is more specific and clearer.

F. It is very difficult to understand the “Finite Projective Plane” on p. 16 without seeing all the details in Suppl., so the authors need to further elaborate it (by moving some contents in Suppl. to main manuscript).

We have now made our section about the finite projective plane construction more self-contained. We have complemented it with the missing information that

was previously in our supplementary material. The supplementary section now contains only the calculation of the witness set size for the perturbed scenarios.

Minor comments

1. Figure 3, y-axis needs in the right bar graph.
2. p.10 line 256, “right size” should be “right side”.
3. Figure 7, $w(0)$ should be $w(1)$?
4. The upper bound of the first order approximation, $2\log_2(r)$ needs to be indicated in Figure 7.

We thank the reviewer for noticing these typos. They are corrected in the current manuscript.

Reviewer #2
(Expertise: Boolean GRNs):

This manuscript analyzes the attractor (phenotype) controllability of 51 Boolean models of biological networks using a previously developed control kernel method. The analysis yields that the average number of nodes that must be controlled to drive the system into one of its natural attractors scales logarithmically with the number of original attractors. This is a mild increase, which is good news for practical applications. Unfortunately, the analysis is not as thorough as it should be and important comparisons are missing. I will elaborate below.

1. Early in the introduction the manuscript states “Common approaches to finding interventions to steer a complex system from an undesirable to a desirable state currently rely on heuristic genetic and greedy algorithms to bound the size of minimal interventions.” This is not a correct assessment. In fact, there is powerful and general theory for attractor control that gives a proven upper bound for minimal interventions that can drive a dynamical system into a desirable attractor. This theory was published in two foundational papers in 2013:

Fiedler et al., Dynamics and Control at Feedback Vertex Sets. I: Informative and Determining Nodes in Regulatory Networks, Journal of Dynamics and Differential Equations, volume 25, pages 563–604 (2013)

Atsushi Mochizuki, Bernold Fiedler, Gen Kurosawa, Daisuke Saito, Dynamics and control at feedback vertex sets. II: A faithful monitor to determine the diversity of molecular activities in regulatory networks, Journal of Theoretical Biology, Volume 335, Pages 130-146 (2013)

The original publications focused on networks that do not have source (input) nodes. The theory was extended to such networks in

Jorge G T Zanudo, G Yang, R Albert, Structure- based control of complex networks with nonlinear dynamics. Proceedings of the National Academy of Sciences of the United States of America, 114(28) (2017). This article also describes attractor control of Boolean systems and demonstrates that for any specific model the number of control nodes is smaller than the upper bound given by Feedback Vertex Set theory. This paper is cited as reference 4 in the manuscript. It is not clear why the authors, having read this article, chose not to disclose that an upper limit to the control nodes already exists: it is made up by all source nodes plus the feedback vertex set of the network.

As mentioned in our response to Reviewer 1, we were remiss in our original submission in not better explaining the relationship between our results and those of feedback vertex set theory and other related control theories. In the revised Introduction, we now compare our control definition with the broader set found in the literature, including the references mentioned by the reviewer:

“Existing literature on Boolean networks contains a number of related definitions of minimal control sets and efficient approaches to finding them. First, stable motif analysis similarly defines control in terms of nodes needed to force the system into particular steady states, which can be found efficiently by intelligent partitioning of the network structure []. Second, defining a control set as a single set of nodes that can force the system to any of the original steady states, feedback vertex sets efficiently produce an upper bound on minimal control set size that does not require knowing the specific dynamics governing each node []. Finally, the method of differentially expressed positive circuits begins with minimal information about the network structure to efficiently find sets of nodes that, when forced, move the system from one of the network’s attractors to another of its attractors [].”

We also explicitly compare sizes of control sets for the biological networks we study in the two most analogous cases (those that aim to force all initial conditions to particular existing attractors): feedback vertex sets and stable motif

analysis. These results can be found in the new Figure 2 and the new section titled “Comparing our results with those obtained using alternative methods”.

Feedback vertex set theory does indeed give a simple upper bound to the size of our control kernel. As seen in the new Figure 2D, the feedback vertex set bound sometimes approaches the size of our minimal control kernels, but in many cases does not, obscuring the logarithmic scaling that forms our main result.

2. The manuscript uses the control kernel method introduced by Kim et al in 2013. This is one of multiple methods available for attractor control of Boolean networks. For example, another method is based on control of stable motifs, as described in Jorge G T Zanudo & R Albert, Cell fate reprogramming by control of intracellular network dynamics, PLOS Computational Biology 11(4): e1004193 (2015). The manuscript needs to support the results by employing multiple methods.

We thank the reviewer for bringing the stable motif analysis technique to our attention, which is indeed closely related to ours. As shown in the new Figure 2B and described in the new section “Comparing our results with those obtained using alternative methods”, we have now explicitly checked that stable motif analysis produces analogous results in terms of scaling of the size of control sets. The stable motif approach differs from ours in the way that it deals with cycles, which are not enumerated explicitly as in our approach but grouped into “quasi-attractors”. Running the open-source stable motif analysis code, we were able to identify minimal control sets for all quasi-attractors in 36 of the biological networks. Analogous to our results that include all synchronous attractors, the mean number of nodes needed to control for individual quasi-attractors also closely follows the logarithm of the total number of quasi-attractors.

3. The identification of the number of attractors is a key component of the work. The authors seem not to be aware of the fact that a multitude of methods exist to identify attractors in Boolean networks. As an example, the article describing the CoLoMoTo Interactive Notebook (Naldi et al. The The CoLoMoTo Interactive

Notebook: Accessible and Reproducible Computational Analyses for Qualitative Biological Networks, *Frontiers in Physiology* 2018) describes some of these methods and their implementation. The results in terms of the number of attractors need to be verified by another method.

We agree that our results depend crucially on correctly identifying the number of attractors in each network. Though we did not state this explicitly in the original manuscript, we extensively tested our code to ensure it correctly finds all attractors. In particular, in cases of relatively small networks, it is possible to check the results of our modular approach with a more direct approach that simply finds all closed loops in the explicitly enumerated state transition graph. For all such cases, consisting of 22 biological and 375 random networks, we have checked that the number of attractors matches exactly using the two techniques. In addition, we have now checked that the biological cases analyzed using stable motif analysis (using code written in a different lab) have the same number of attractors in the 15 cases of networks we were able to analyze that have only fixed point attractors (where the definitions of our attractors and stable motif's quasi-attractors match).

We have now added a version of the above text as the Supplemental Material section "Validating attractors".

As a side note, we thank the reviewer for mentioning CoLoMoTo, which we were not previously aware of and may become useful for future work.

4. Another point related to the number of attractors. It is well documented that a large subset of cyclic attractors of synchronous Boolean models are artifacts: they rely on the synchronicity of node state changes and are destroyed for any perturbations to that synchronicity. The number of attractors decreases dramatically in asynchronous networks (see for example Greil F, Drossel B. Dynamics of critical Kauffman networks under asynchronous stochastic update. *Phys Rev Lett.* 2005;95: 048701.). The number of synchronous attractors is not a very good reflection of the phenotype diversity of biological systems. Thus, the

identified logarithmic scaling may be due to an inflation of the number of attractors.

The issue of synchronous versus asynchronous updating is important to understanding implications of our results, particularly for larger networks than those we have studied here. We had previously included a brief discussion of the effects of asynchronous updating in the supplemental material, but we agree that this issue is important enough to expand our discussion there and mention it more prominently in the main text.

Most important to mention here is that our logarithmic scaling result stands even when excluding all networks that have any cycles, leaving 24 of the 49 networks that we now specifically highlight in Fig. 13F. In these cases, all attractors are fixed points, and attractors and control kernels are equivalent given any updating scheme.

In addition, the fact that the number of attractors after pinning distinguishing nodes is typically small (see new Fig. 5) leads us to conclude that an inflation of the number of attractors due to unstable cycles is not an issue in the networks we study. We do anticipate that a proliferation of synchronous cycles could become a problem for much larger networks, but we do not study those here. We now mention this explicitly in the “cyclic attractors” sub-section of the “Theoretical Interpretation” section:

“A proliferation of cycles created during the intermediate rounds of pinning could potentially prevent our iterative procedure from converging quickly. Nonetheless, this is not something we observe in the biological networks we analyzed. ... Whether biological networks much larger than the sizes we analyzed display a proliferation of cycles similar to RBNs is something we cannot currently test. Indeed, it is an open question whether networks displaying large numbers of cycles are relevant to biology [] and complexity science more generally. It is also worth noting that many cycles in large

networks become unstable when the network state is updated in a nondeterministic asynchronous way [] (see Supplemental Material).”

We have also expanded our discussion in the Supplemental Material to make clear that, while we view a generalization to asynchronous updating to be beyond the scope of this work, we do not have a reason to believe that it would fundamentally change the result for the networks we test. This is because we expect any loss of cyclic attractors in the asynchronous case would cause a corresponding decrease in the size of witness sets and therefore CKs (see the updated discussion in the Supplementary Material section titled “The effects of asynchronous updating”).

Reviewers' Comments:

Reviewer #1:

Remarks to the Author:

I appreciate that the authors tried to address the previous comments of this reviewer. The revised manuscript has been improved significantly, but I still have some concerns as follows.

Major comments:

In my previous comments, I asked the relationship between distinguishing nodes and those obtained by alternative methods, which seems not properly addressed. I am wondering how much those sets are overlapped (not simply the analysis of size).

And, I also asked to denote $2\log_2(r)$ in Fig. 8. I am still curious why it is bounded by $\log_2(r)$ instead of $2\log_2(r)$. May I assume that $\log_2(r)$ includes not only distinguishing nodes but also input nodes?

Minor comments:

1. Line #214: "deleting column x_j on each side" should be "deleting column x_j and x'_j on each side"?
2. Line #287: "final state" should be "next state"?
3. In line #395, r_j is defined as "the number of attractors given the input state corresponding to attractor j ", but in line #564, r_j is defined as "the number of attractors sharing the j -th input configuration"? It seems there is a mistake caused by this confusion in line #397, $r = \sum(r_j)$ where r_j is wrong according to its definition of line #395. For instance, if there are four attractors and two of them share input states then $r_1 = r_2 = r_3 = r_4 = 2$, $\sum(r_j) = 2+2+2+2=8$, so it contradicts that the total number of attractors is 4. This means we have to follow the definition of line #564. This confusion needs to be clarified and the definition of $w_j(1)$ should be described accordingly.

Reviewer #2:

Remarks to the Author:

The manuscript has improved a lot by the revision and additional analyses. I would like the authors to consider a few remaining suggestions.

1. The feedback vertex set should be defined; not all readers will be familiar with it.
2. It is interesting that all three control methods referred to in the Introduction: stable motifs, feedback vertex sets and differentially expressed positive circuits, rely on controlling certain circuits (in the graph theoretical sense). This fact does not come across in the paragraph. The paragraph should be revised so it better illuminates the ideas of these three methods. For example, instead of writing "intelligent partitioning of the network structure" in the sentence about stable motif control, a more insightful description would be "identification of positive circuits in the network that can self-sustain an associated state". Or, with a more technical term, "positive circuits in the network that corresponds to trap spaces in the dynamics".
3. Also related to the previous point, it may be interesting to consider in the Discussion in what way the control kernel reflects the circuits of the network.
4. The manuscript states "the method of stable motifs groups any cyclic attractors into quasi-attractors", which gives the impression that all cyclic attractors are grouped into a single quasi-

attractor. In fact, only those cyclic attractors whose stationary parts are identical (i.e. cyclic attractors that reside in the same trap space) are grouped into a quasi-attractor.

Reviewer #1 (Remarks to the Author):

I appreciate that the authors tried to address the previous comments of this reviewer. The revised manuscript has been improved significantly, but I still have some concerns as follows.

Major comments:

In my previous comments, I asked the relationship between distinguishing nodes and those obtained by alternative methods, which seems not properly addressed. I am wondering how much those sets are overlapped (not simply the analysis of size).

As the focus of our result is on the relation between the number of controlling nodes and the number of attractors, we had initially ignored the identity of the controlling nodes. Such analysis becomes complicated by the fact that many different controlling node sets of the same size may exist for each attractor. This ambiguity means that, if we were to find little overlap between control sets produced by different methods, this would not necessarily imply that the methods are fundamentally distinct. Furthermore, attempting to find all possible minimal control sets for every attractor becomes computationally infeasible.

Even so, we agree with the reviewer that a simple comparison of the particular control sets found by each method is useful, especially in light of concrete applications. We have therefore expanded the section comparing our results to those obtained using feedback vertex sets and stable motifs. In both cases the overlap between our sets and those found with the alternative methods is large:

Comparing control kernels to stable motif control sets (for fixed points only), the average overlap across all 4262 attractors is 94%. When comparing union control kernels to feedback vertex sets, the average proportion of control kernels that are part of one particular feedback vertex set is 88%.

More details can be found in a supplemental figure (new Fig. 17) with the numerical details and the metrics that we use for the comparison.

And, I also asked to denote $2\log_2(r)$ in Fig. 8. I am still curious why it is bounded by $\log_2(r)$ instead of $2\log_2(r)$. May I assume that $\log_2(r)$ includes not only distinguishing nodes but also input nodes?

To simplify the discussion we had opted to highlight the conservative bound of $2\log_2(r)$. Our focus was more on the logarithmic scaling than the slope of the scaling. We have now complemented the corresponding section in the Methods with the details that show that, in typical situations in which different settings of the input nodes produce similar numbers of attractors, the average control kernel size is rarely larger than $\log_2(r)$.

To answer the reviewer's specific question, $\log_2(r)$ does include the input nodes, but whenever the attractors are (approximately) evenly distributed over the set of initial input configurations, the size of the first order approximation of the CK is not very sensitive to the number m of input nodes. This is the typical behavior in all the ensembles we consider (both biological and random networks), causing $\langle |CK^{(1)}| \rangle$ to be much closer to $\log_2(r)$ than $2\log_2(r)$.

Therefore, while $2\log_2(r)$ is a simpler, conservative upper bound on $\langle |CK^{(1)}| \rangle$, the slope of the scaling law line is effectively 1.

To clarify this point, we expanded the Methods section "Upper bound on the first order approximation of CK size", calling the bound on $\langle |CK^{(1)}| \rangle$ "a simple, conservative upper bound ... that depends only on r " and pointing out that "We find empirically that the actual $\langle |CK^{(1)}| \rangle$ values lie significantly below this conservative bound." We also added a more detailed bound that depends explicitly on m and a quantity we call the "input entropy" μ (new Eqn. (9)), and we show that the m and μ terms nearly exactly cancel each other in the biological cases we test (new Fig. 11).

Finally, we note:

With the larger statistics provided by the ensembles of random networks, we find a few cases in which $\langle |CK^{(1)}| \rangle$ is slightly greater than $\log_2(r)$ (four cases in the *zero threshold* ensemble and four more cases in the *balanced threshold* ensemble; see Fig. 7B), but still never greater than $2\log_2(r)$, as expected.

Minor comments:

1. Line #214: "deleting column x_j on each side" should be "deleting column x_j and x'_j on each side"?

We agree with this suggestion, and have changed the text accordingly.

2. Line #287: “final state” should be “next state”?

We agree with this as well. The text is now updated.

3. In line #395, r_j is defined as “the number of attractors given the input state corresponding to attractor j ”, but in line #564, r_j is defined as “the number of attractors sharing the j -th input configuration”? It seems there is a mistake caused by this confusion in line #397, $r = \sum(r_j)$ where r_j is wrong according to its definition of line #395. For instance, if there are four attractors and two of them share input states then $r_1 = r_2 = r_3 = r_4 = 2$, $\sum(r_j) = 2+2+2+2=8$, so it contradicts that the total number of attractors is 4. This means we have to follow the definition of line #564. This confusion needs to be clarified and the definition of $w_j(1)$ should be described accordingly.

We thank the reviewer for noticing this. What we previously wrote in line #395 is not what we meant. The correct definition of r_j is “the number of attractors sharing the j -th input configuration”. All following considerations and formulae were correct and consistent with the definition in the original line #564. Line #395 has been corrected.

Reviewer #2 (Remarks to the Author):

The manuscript has improved a lot by the revision and additional analyses. I would like the authors to consider a few remaining suggestions.

1. The feedback vertex set should be defined; not all readers will be familiar with it.

We agree with the reviewer. Its definition has been added to the Introduction, where it first appears.

2. It is interesting that all three control methods referred to in the Introduction: stable motifs, feedback vertex sets and differentially expressed positive circuits, rely on controlling certain circuits (in the graph theoretical sense). This fact does not come across in the paragraph. The paragraph should be revised so it better illuminates the ideas of these three methods. For example, instead of writing "intelligent partitioning of the network structure" in the sentence about stable motif control, a more insightful description would be "identification of positive circuits in the network that can self-sustain an associated state". Or, with a more technical term, "positive circuits in the network that corresponds to trap spaces in the dynamics".

We have now updated the text to take this suggestion into account.

3. Also related to the previous point, it may be interesting to consider in the Discussion in what way the control kernel reflects the circuits of the network.

We agree. We have added the following paragraph to the Discussion:

Control kernels are fundamentally related to closed circuits in the regulatory networks, as suggested by the close relation to the methods of stable motifs and feedback vertex sets that rely on identifying these closed circuits. In particular, we expect that each non-input control kernel node is part of a directed closed circuit in the regulatory networks (as paths in the network not involved in cycles correspond to deterministic cascades that cannot support multiple possible states).

In addition, as a similar question has also been asked by our other reviewer, we have now extended our section on the comparison between our results and other methods, showing large overlap between control kernels and feedback vertex sets: on average, 88% of nodes that appear in control kernels in each network are part of one particular feedback vertex set (see also new supplemental Fig. 17).

4. The manuscript states "the method of stable motifs groups any cyclic attractors into quasi-attractors", which gives the impression that all cyclic attractors are grouped into a single quasi-attractor. In fact, only those cyclic attractors whose stationary parts are identical (i.e. cyclic attractors that reside in the same trap space) are grouped into a quasi-attractor.

We agree with this suggestion. We have also removed this ambiguity from the footnote in the Introduction where we refer to stable motifs.

Reviewers' Comments:

Reviewer #1:

Remarks to the Author:

All of my comments were properly addressed and I am very happy to recommend its publication.

(One minor point: "final state" in line #300 should be corrected as "next state".)